# Beyond spectral gap:
# The role of the topology in decentralized learning

**Thijs Vogels**[*]
EPFL

**Hadrien Hendrikx**[*]
EPFL

**Martin Jaggi**
EPFL

## Abstract

In data-parallel optimization of machine learning models, workers collaborate to improve their estimates of the model: more accurate gradients allow them to use larger learning rates and optimize faster. We consider the setting in which all workers sample from the same dataset, and communicate over a sparse graph (decentralized). In this setting, current theory fails to capture important aspects of real-world behavior. First, the 'spectral gap' of the communication graph is not predictive of its empirical performance in (deep) learning. Second, current theory does not explain that collaboration enables *larger* learning rates than training alone. In fact, it prescribes *smaller* learning rates, which further decrease as graphs become larger, failing to explain convergence in infinite graphs. This paper aims to paint an accurate picture of sparsely-connected distributed optimization when workers share the same data distribution. We quantify how the graph topology influences convergence in a quadratic toy problem and provide theoretical results for general smooth and (strongly) convex objectives. Our theory matches empirical observations in deep learning, and accurately describes the relative merits of different graph topologies. Code: `github.com/epfml/topology-in-decentralized-learning`

## 1 Introduction

Distributed data-parallel optimization algorithms help us tackle the increasing complexity of machine learning models and of the data on which they are trained. We can classify those training algorithms as either *centralized* or *decentralized*, and we often consider those settings to have different benefits over training 'alone'. In the *centralized* setting, workers compute gradients on independent mini-batches of data, and they average those gradients between all workers. The resulting lower variance in the updates enables larger learning rates and faster training. In the *decentralized* setting, workers average their models with only a sparse set of 'neighbors' in a graph instead of all-to-all, and they may have private datasets sampled from different distributions. As the benefit of decentralized learning, we usually focus only on the (indirect) access to other worker's datasets, and not of faster training.

While decentralized learning is typically studied with heterogeneous datasets across workers, sparse (decentralized) averaging between is also useful when worker's data is identically distributed (i.i.d.) [15]. As an example, sparse averaging is used in data centers to mitigate communication bottlenecks [1]. In fact the D-SGD algorithm [11], on which we focus in this work, performs well mainly in this setting, while algorithmic modifications [14, 21, 22] are required to yield good performance on heterogeneous objectives. When the data is i.i.d., the goal of sparse averaging is to optimize faster, just like in centralized (all-to-all) graphs.

Yet, current decentralized learning theory poorly explains the i.i.d. case. Analyses typically show that, for *small enough* learning rates, training with sparse averaging behaves the same as with all-to-all averaging [11, 8]. Compared to training alone with the *same small learning rate*, all-to-all averaging

---

[*]Equal contribution. Corresponding authors `thijs.vogels@epfl.ch` and `hadrien.hendrikx@epfl.ch`.

36th Conference on Neural Information Processing Systems (NeurIPS 2022).

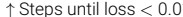

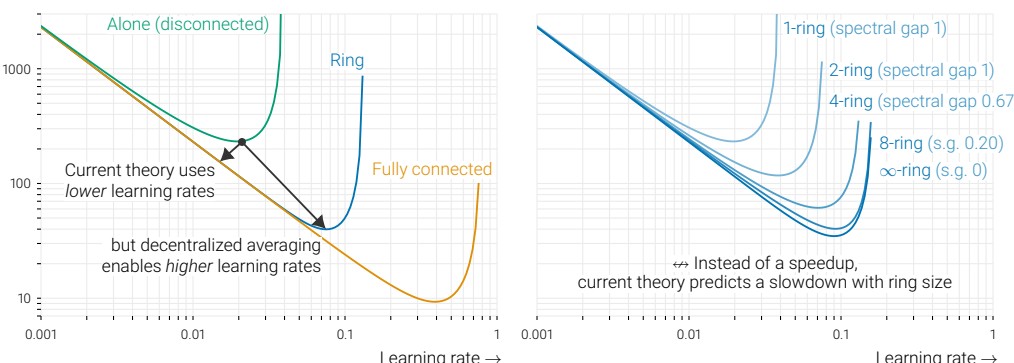

Figure 1: 'Time to target' for D-SGD [11] with constant learning rates on an i.i.d. isotropic quadratic dataset (Section 3). The noise disappears at the optimum. Compared to optimizing alone, 32 workers in a ring (*left*) are faster for any learning rate, but the largest improvement comes from being able to use a large learning rate. This benefit is not captured by current theory, which prescribes a smaller learning rate than training alone. On the *right*, we see that rings of increasing size enable larger learning rates and faster optimization. Because a ring's spectral gap goes to zero with the size, this cannot be explained by current theory.

reduces the gradient variance by the number of workers. In practice, however, such small learning rates would never be used. In fact, a reduction in variance should allow us to use a *larger* learning rate than training alone, rather than imposing a *smaller* one. Contrary to current theory, we show that averaging reduces the variance from the start, instead of just asymptotically. Lower variance increases the maximum learning rate, which directly speeds up convergence. We characterize how much averaging with various communication graphs reduces the variance, and show that centralized performance is not always achieved when using optimal large learning rates. The behavior we explain is illustrated in Figure 1.

In current convergence rates, the graph topology appears through the *spectral gap* of its averaging (gossip) matrix. The spectral gap poses a conservative lower bound on how much one averaging step brings all worker's models closer together. The larger, the better. If the spectral gap is small, a significantly smaller learning rate is required to make the algorithm behave close to SGD with all-to-all averaging with the same learning rate. Unfortunately, we experimentally observe that, both in deep learning and in convex optimization, the spectral gap of the communication graph is *not predictive* of its performance under realistically tuned learning rates.

The problem with the spectral gap quantity is clearly illustrated in a simple example. Let the communication graph be a ring of varying size. As the size of the ring increases to infinity, its spectral gap goes to zero, since it becomes harder and harder to achieve consensus between all the workers. This leads to the optimization progress predicted by current theory to go to zero as well. Yet, this behavior does not match the empirical behavior of the rings with i.i.d. data. As the size of the ring increases, the convergence rate actually *improves* (Figure 1), until it saturates at a point that depends on the problem.

In this work, we aim to accurately describe the behavior of i.i.d. distributed learning algorithms with sparse averaging, both in theory and in practice. We quantify the role of the graph in a quadratic toy problem designed to mimic the initial phase of deep learning (Section 3), showing that averaging enables a larger learning rate. From these insights, we derive a problem-independent notion of 'effective number of neighbors' in a graph that is consistent with time-varying topologies and infinite graphs, and is predictive of a graph's empirical performance in both convex and deep learning. We provide convergence proofs for convex and (strongly) convex objectives that only mildly depend on the spectral gap of the graph (Section 4), and consider the whole spectrum instead. At its core, our analysis does not enforce global consensus, but only between workers that are close to each other in the graph. Our theory shows that sparse averaging provably enables larger learning rates and thus speeds up optimization. These insights prove to be relevant in deep learning, where we accurately describe the performance of a variety of topologies, while their spectral gap does not (Section 5).

## 2 Related work

**Decentralized SGD**   This paper studies decentralized SGD. Koloskova et al. [8] obtain the tightest bounds for this algorithm in the general setting where workers optimize heterogeneous objectives. Contrary to their work, we focus primarily on the case where all workers sample i.i.d. data from the same distribution. This important case is not described in a meaningful way by their analysis: while they show that gossip averaging reduces the asymptotic variance suffered by the algorithm, the fast initial linear decrease term in their convergence rate depends on the spectral gap of the gossip matrix. This key term does not improve through collaboration and gives rise to a *smaller learning rate* than training alone. Besides, as discussed above, this implies that optimization is not possible in the limit of large graphs, even in the absence of heterogeneity: for instance, the spectral gap of an infinite ring is zero, which would lead to a learning rate of zero as well.

These rates suggest that decentralized averaging speeds up the last part of training (dominated by variance), at the cost of slowing down the initial (linear convergence) phase. Beyond the work of Koloskova et al. [8], many papers focus on *linear speedup* (in the variance phase) over optimizing alone, and prove similar results in a variety of settings [11, 20, 12]. All these results rely on the following insight: while linear speedup is only achieved for small learning rates, SGD eventually requires such small learning rates anyway (because of, e.g., variance, or non-smoothness). This observation leads these works to argue that "topology does not matter". This is the case indeed, but only for very small learning rates, as shown in Figure 1. In practice, averaging speeds up both the initial *and* last part of training. This is what we show in this work, both in theory and in practice.

Another line of work studies D-(S)GD under statistical assumptions on the local data. In particular, Richards and Rebeschini [18] show favorable properties for D-SGD with graph-dependent implicit regularization and attain optimal statistical rates. Their suggested learning rate does depend on the spectral gap of the communication network, and it goes to zero when the spectral gap shrinks. Richards and Rebeschini [17] also show that larger (constant) learning rates can be used in decentralized GD, but their analysis focuses on decentralized kernel regression. It does not cover stochastic gradients, and relies on statistical concentration of local objectives rather than analysis on local neighborhoods.

**Gossiping in infinite graphs**   An important feature of our results is that they only mildly depend on the spectral gap, and so they apply independently of the size of the graph. Berthier et al. [3] study acceleration of gossip averaging in infinite graphs, and obtain the same conclusions as we do: although spectral gap is useful for asymptotics, it fails to accurately describe the transient regime of averaging. This is especially limiting for optimization (compared to of just averaging), as new local updates need to be averaged at every step. The transient regime of averaging deeply matters. Indeed, it impacts the quality of the gradient updates, and so it rules the asymptotic regime of optimization.

**The impact of the topology**   Some works on linear speedup [11] argue that the topology of the graph does not matter. This is only true for asymptotic rates in specific settings, as illustrated in Figure 1. Neglia et al. [16] investigate the impact of the topology on decentralized optimization, and contradict this claim. Compared to us, they make different noise assumptions, which in particular depend on the spectral distribution of the noise over the eigenvalues of the Laplacian (thus mixing computation and communication aspects). Although they show that the topology has an impact in the early phases of training (just like we do), they still get an unavoidable dependence on the spectral gap of the graph. Our results are different in nature, and show the benefits of averaging and the impact of the topology through the choice of large learning rates.

Another line of work studies the interaction of topology with particular patterns of data heterogeneity [4, 2], and how to optimize graphs with this heterogeneity in mind. These works "only" show a benefit from one-step gossip averaging and this is thus what they optimize the graph for. In contrast, we show that it is possible to benefit from distant workers beyond direct neighbors, too. This is an orthogonal direction, though the insights from our work could be used to strengthen their results.

**Time-varying topologies**   Time-varying topologies are popular for decentralized deep learning in data centers due to their strong mixing [1, 23]. The benefit of varying the communication topology over time is not easily explained through standard theory, but requires dedicated analysis [25]. While our proofs only cover static topologies, the quantities that appear in our analysis can be computed for time-varying schemes, too. With these quantities, we can empirically study static and time-varying schemes in the same framework.

# 3 A toy problem: D-SGD on isotropic random quadratics

Before analyzing decentralized stochastic optimization through theory for general convex objectives and deep learning experiments, we first investigate a simple toy example that illustrates the behavior we want to explain in the analysis. In this setting, we can exactly characterize the convergence of decentralized SGD. We also introduce concepts that will be used throughout the paper.

We consider $n$ workers that jointly optimize an isotropic quadratic $\mathbb{E}_{\mathbf{d} \sim \mathcal{N}^d(0,1)} \frac{1}{2}(\mathbf{d}^\top \mathbf{x})^2 = \frac{1}{2}\|\mathbf{x}\|^2$ with a unique global minimum $\mathbf{x}^\star = \mathbf{0}$. The workers access the quadratic through stochastic gradients of the form $\mathbf{g}(\mathbf{x}) = \mathbf{d}\mathbf{d}^\top \mathbf{x}$, with $\mathbf{d} \sim \mathcal{N}^d(0,1)$. This corresponds to a linear model with infinite data, and where the model can fit the data perfectly, so that stochastic noise goes to zero close to the optimum. We empirically find that this simple model is a meaningful proxy for the initial phase of (over-parameterized) deep learning (Section 5). A benefit of this model is that we can compute exact rates for it. These rates illustrate the behavior that we capture more generally in the theory of Section 4. Appendix C contains a detailed version of this section that includes full derivations.

The stochasticity in this toy problem can be quantified by the *noise level*

$$\zeta = \sup_{\mathbf{x} \in \mathbb{R}^d} \frac{\mathbb{E}_{\mathbf{d}}\|\mathbf{d}\mathbf{d}^\top \mathbf{x}\|^2}{\|\mathbf{x}\|^2}, \tag{1}$$

which is equal to $\zeta = d + 2$, due to the random normal distribution of $\mathbf{d}$.

The workers run the D-SGD algorithm [11]. Each worker $i$ has its own copy $\mathbf{x}_i \in \mathbb{R}^d$ of the model, and they alternate between local model updates $\mathbf{x}_i \leftarrow \mathbf{x}_i - \eta \mathbf{g}(\mathbf{x}_i)$ and averaging their models with others: $\mathbf{x}_i \leftarrow \sum_{j=1}^n w_{ij}\mathbf{x}_j$. The averaging weights $w_{ij}$ are summarized in the *gossip matrix* $\mathbf{W} \in \mathbb{R}^{n \times n}$. A non-zero weight $w_{ij}$ indicates that $i$ and $j$ are directly connected. In the following, we assume that $\mathbf{W}$ is symmetric and doubly stochastic: $\sum_{j=1}^n w_{ij} = 1 \; \forall i$.

On our objective, D-SGD either converges or diverges linearly. Whenever it converges, i.e. when the learning rate is small enough, there is a convergence rate $r$ such that

$$\mathbb{E}\|\mathbf{x}_i^{(t)}\|^2 \leq (1-r)\|\mathbf{x}_i^{(t-1)}\|^2,$$

with equality as $t \to \infty$ (proofs in Appendix C). When the workers train alone ($\mathbf{W} = \mathbf{I}$), the convergence rate for a given learning rate $\eta$ reads:

$$r_{\text{alone}} = 1 - (1-\eta)^2 - (\zeta-1)\eta^2. \tag{2}$$

The optimal learning rate $\eta^\star = \frac{1}{\zeta}$ balances the optimization term $(1-\eta)^2$ and the stochastic term $(\zeta-1)\eta^2$. In the centralized (fully connected) setting ($w_{ij} = \frac{1}{n} \; \forall i, j$), the rate is simple as well:

$$r_{\text{centralized}} = 1 - (1-\eta)^2 - \frac{(\zeta-1)\eta^2}{n}. \tag{3}$$

Averaging between $n$ workers reduces the impact of the gradient noise, and the optimal learning rate grows to $\eta^\star = \frac{n}{n+\zeta-1}$. D-SGD with a general gossip matrix $\mathbf{W}$ interpolates those results.

To quantify the reduction of the $(\zeta-1)\eta^2$ term in general, we introduce the *problem-independent* notion of *effective number of neighbors* $n_{\mathbf{W}}(\gamma)$ of the gossip matrix $\mathbf{W}$ and *decay parameter* $\gamma$.

**Definition A.** The effective number of neighbors $n_{\mathbf{W}}(\gamma) = \lim_{t \to \infty} \frac{\sum_{i=1}^n \text{Var}[\mathbf{y}_i^{(t)}]}{\sum_{i=1}^n \text{Var}[\mathbf{z}_i^{(t)}]}$ measures the ratio of the asymptotic variance of the processes

$$\mathbf{y}^{(t+1)} = \sqrt{\gamma} \cdot \mathbf{y}^{(t)} + \boldsymbol{\xi}^{(t)}, \quad \text{where } \mathbf{y}^{(t)} \in \mathbb{R}^n \text{ and } \boldsymbol{\xi}^{(t)} \sim \mathcal{N}^n(0,1) \tag{4}$$

and

$$\mathbf{z}^{(t+1)} = \mathbf{W}(\sqrt{\gamma} \cdot \mathbf{z}^{(t)} + \boldsymbol{\xi}^{(t)}), \quad \text{where } \mathbf{z}^{(t)} \in \mathbb{R}^n \text{ and } \boldsymbol{\xi}^{(t)} \sim \mathcal{N}^n(0,1). \tag{5}$$

We call $\mathbf{y}$ and $\mathbf{z}$ *random walks* because workers repeatedly add noise to their state, somewhat like SGD's parameter updates. This should not be confused with a 'random walk' over nodes in the graph.

Since averaging with $\mathbf{W}$ decreases the variance of the random walk by at most $n$, the effective number of neighbors is a number between $1$ and $n$. The decay $\gamma$ modulates the sensitivity to communication delays. If $\gamma = 0$, workers only benefit from averaging with their direct neighbors. As $\gamma$ increases,

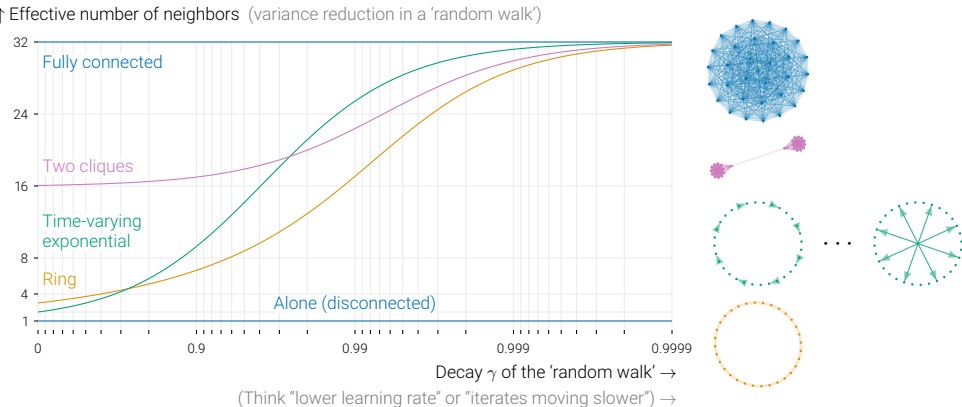

Figure 2: The effective number of neighbors for several topologies (Appendix B) measured by their variance reduction in Equation 5. The point $\gamma$ on the $x$-axis that matters depends on the learning rate and the task. Which topology is 'best' varies from problem to problem. For large decay rates $\gamma$ (corresponding small learning rates), all connected topologies achieve variance reduction close to a fully connected graph. For small decay rates (large learning rates), workers only benefit from their direct neighbors (e.g. 3 in a ring). These curves can be computed explicitly for constant topologies, and simulated efficiently for the time-varying exponential scheme [1].

multi-hop connections play an increasingly important role. As $\gamma$ approaches 1, delayed and undelayed noise contributions become equally weighted, and the reduction tends to $n$ for any connected topology.

For regular doubly-stochastic symmetric gossip matrices $\mathbf{W}$ with eigenvalues $\lambda_1, \ldots, \lambda_n$, $n_\mathbf{W}(\gamma)$ has a closed-form expression

$$n_\mathbf{W}(\gamma) = \frac{\frac{1}{1-\gamma}}{\frac{1}{n} \sum_{i=1}^{n} \frac{\lambda_i{}^2}{1-\lambda_i^2 \gamma}}. \tag{6}$$

The notion of variance reduction in random walks, however, naturally extends to infinite topologies or time-varying averaging schemes as well. Figure 2 illustrates $n_\mathbf{W}$ for various topologies.

In our exact characterization of the convergence of D-SGD on the isotropic quadratic toy problem (Appendix C), we find that the effective number of neighbors appears in place of the number of workers $n$ in the fully-connected rate of Equation 3. The rate is the unique solution to

$$r = 1 - (1 - \eta)^2 - \frac{(\zeta - 1)\eta^2}{n_\mathbf{W}\left(\frac{(1-\eta)^2}{1-r}\right)}. \tag{7}$$

For fully-connected and disconnected $\mathbf{W}$, $n_\mathbf{W}(\gamma) = n$ or 1 respectively, irrespective of $\gamma$, and Equation 7 recovers Equations 2 and 3. For other graphs, the effective number of workers depends on the learning rate. Current theory only considers the case where $n_\mathbf{W} \approx n$, but the small learning rates this requires can make the term $(1 - \eta)^2$ too large, defeating the purpose of collaboration.

Beyond this toy problem, we find that the proposed notion of effective number of neighbors is also meaningful in the analysis of general objectives (Section 4) and in deep learning (Section 5).

## 4 Theoretical analysis

In the previous section, we have derived exact rates for a specific function. Now we present convergence rates for general (strongly) convex functions that are consistent with our observations in the previous section. We obtain rates that depend on the level of noise, the hardness of the objective, and the topology of the graph. We will assume the following randomized model for D-SGD:

$$\mathbf{x}_i^{(t+1)} = \begin{cases} \mathbf{x}_i^{(t)} - \eta \nabla f_{\xi_i^{(t)}}(\mathbf{x}_i^{(t)}) & \text{with probability } \frac{1}{2}, \\ \sum_{j=1}^{n} w_{ij} \mathbf{x}_j^{(t)} & \text{otherwise,} \end{cases} \tag{8}$$

where $f_{\xi_i^{(t)}}$ represent sampled data points and the gossip weights $w_{ij}$ are elements of $\mathbf{W}$. This randomized model yields a clean analysis, but similar results hold for standard D-SGD (Appendix D.4).

**Assumption B.** The stochastic gradients are such that: (I) $\xi_i^{(t)}$ and $\xi_j^{(\ell)}$ are independent for all $t, \ell$ and $i \neq j$. (II) $\mathbb{E}\left[f_{\xi_i^{(t)}}\right] = f$ for all $t, i$ (III) $\mathbb{E}\left\|\nabla f_{\xi_i^{(t)}}(\mathbf{x}^\star)\right\|^2 \leq \sigma^2$ for all $t, i$, where $\mathbf{x}^\star$ is a minimizer of $f$. (IV) $f_{\xi_i^{(t)}}$ is convex and $\zeta$-smooth for all $t, i$. (V) $f$ is $\mu$-strongly-convex for $\mu \geq 0$ and $L$-smooth.

The smoothness $\zeta$ of the stochastic functions $f_\xi$ defines the level of noise in the problem (the lower, the better). The ratio $\zeta/L$ compares the difficulty of optimizing with stochastic gradients to the difficulty with the true global gradient (before reaching the 'variance region' of distance $\mathcal{O}(\sigma^2)$ to the optimum). Assuming better smoothness for the global average objective than for the local functions is key to showing the benefit of averaging between workers. Without communication, convergence to the variance region is ensured for learning rates $\eta \leq 1/\zeta$. If $\zeta \approx L$, there is little noise and cooperation does not help before $\|\mathbf{x}^{(t)} - \mathbf{x}^\star\|^2 \approx \sigma^2$. Yet, in noisy regimes ($\zeta \gg L$), such as in Section 3 in which $\zeta = d + 2 \gg 1 = L$, averaging enables larger step-sizes up to $\min(1/L, n/\zeta)$, greatly speeding up the initial training phase. This is precisely what we prove in Theorem I.

*If* the workers always remain close ($\mathbf{x}_i \approx \frac{1}{n}(\mathbf{x}_1 + \ldots + \mathbf{x}_n)$ $\forall i$, or equivalently $\frac{1}{n}\mathbf{1}\mathbf{1}^\top \mathbf{x} \approx \mathbf{x}$), D-SGD behaves the same as SGD on the average parameter $\frac{1}{n}\sum_{i=1}^n \mathbf{x}_i$, and the learning rate depends on $\max(\zeta/n, L)$, showing a reduction of variance by $n$. To maintain "$\frac{1}{n}\mathbf{1}\mathbf{1}^\top \mathbf{x} \approx \mathbf{x}$", however, we require a small learning rate. This is a common starting point for the analysis of D-SGD, in particular for the proofs in Koloskova et al. [8]. On the other extreme, if we do not assume closeness between workers, "$\mathbf{I}\mathbf{x} \approx \mathbf{x}$" always holds. In this case, there is no variance reduction, but no requirement for a small learning rate either. In Section 3, we found that, at the optimal learning rate, workers are *not* close to all other workers, but they *are* close to others that are not too far away in the graph.

We capture the concept of 'local closeness' by defining an averaging matrix $\mathbf{M}$. It allows us to consider semi-local averaging beyond direct neighbors, but without fully averaging with the whole graph. We ensure that "$\mathbf{M}\mathbf{x} \approx \mathbf{x}$", leading to some improvement in the smoothness between $\zeta$ and $\zeta/n$, interpolating between the two previous cases. Each matrix $\mathbf{M}$ implies a requirement on the learning rate, as well as an improvement in smoothness. Based on Section 3, we therefore focus on a specific family of matrices that strike a good balance between the two: We choose $\mathbf{M}$ as the covariance of a decay-$\gamma$ 'random walk process' with the graph, meaning that

$$\mathbf{M} = (1 - \gamma)\sum_{k=1}^\infty \gamma^{k-1}\mathbf{W}^{2k} = (1 - \gamma)\mathbf{W}^2(1 - \gamma\mathbf{W}^2)^{-1}. \tag{9}$$

Varying $\gamma$ induces a spectrum of averaging neighborhoods from $\mathbf{M} = \mathbf{W}^2$ ($\gamma = 0$) to $\mathbf{M} = \frac{1}{n}\mathbf{1}\mathbf{1}^\top$ ($\gamma = 1$). $\gamma$ also implies an effective number of neighbors $n_\mathbf{W}(\gamma)$: the larger $\gamma$, the larger $n_\mathbf{W}(\gamma)$.

Theorem I provides convergence rates for any value of $\gamma$, but the best rates are obtained for a specific $\gamma$ that balances the benefit of averaging with the constraint it imposes on closeness between neighbors. In the following theorem, we assume that $\mathbf{M}_{ii} = \mathbf{M}_{jj}$ for all $i, j$, so that $\mathbf{M}_{ii}^{-1} = n_\mathbf{W}(\gamma)$: the effective number of neighbors defined in (6) is equal to the inverse of the self-weights of matrix $\mathbf{M}$. Otherwise, all results hold by replacing $n_\mathbf{W}(\gamma)$ with $\min_i \mathbf{M}_{ii}^{-1}$.

**Theorem I.** If Assumption B holds, and the learning rate satisfies

$$\eta \leq \min\left(\frac{1}{8(\zeta/n_\mathbf{W}(\gamma) + L)}, \frac{1 - \gamma\lambda_2(\mathbf{W})}{2n_\mathbf{W}(\gamma)L}\right), \tag{10}$$

then the iterates obtained by (8) verify

$$\|\mathbf{x}^{(t)} - \mathbf{x}^\star\|_\mathbf{M}^2 + \frac{1}{n_\mathbf{W}(\gamma)}\|\mathbf{x}^{(t)}\|_{\mathbf{I}-\mathbf{M}}^2 \leq \left(1 - \frac{\eta\mu}{2}\right)^t C_0 + \frac{8\eta\sigma^2}{n_\mathbf{W}(\gamma)}, \tag{11}$$

The bound on the learning rate (10) represents the tension between (I) reducing the noise $\zeta$ by averaging with more people (larger $n_\mathbf{W}(\gamma)$), which is the first term in the minimum, and (II) staying close to all of them. A large spectral gap $1 - \lambda_2(\mathbf{W})$ reduces the second constraint, but we allow non-trivial learning rates $\eta > 0$ even when $\lambda_2(\mathbf{W}) = 1$ (infinite graphs) as long as $\gamma < 1$.

Theorem I gives a rate for each parameter $\gamma$ that controls the local neighborhood size. The task that remains is to find the $\gamma$ parameter that gives the best convergence guarantees (the largest learning rate). As explained before, one should never reduce the learning rate in order to be close to others,

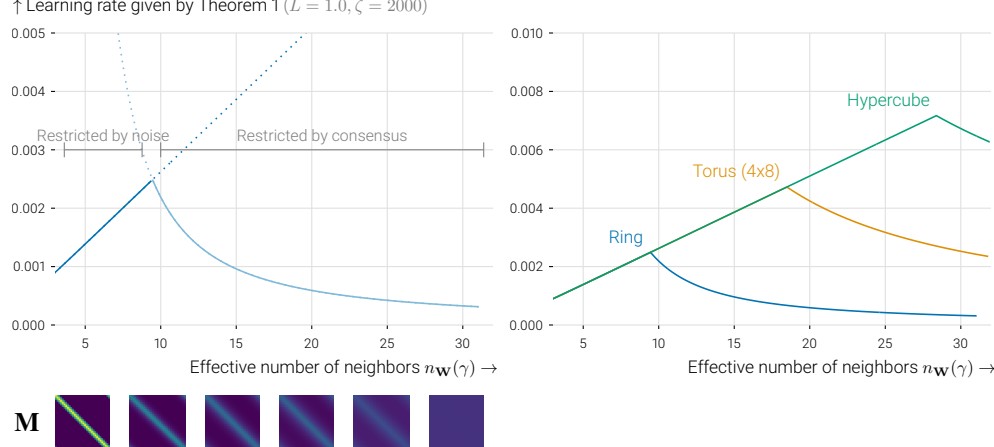

Figure 3: Maximum learning rates prescribed by Theorem I, varying the parameter $\gamma$ that implies an effective neighborhood size ($x$-axis) and an averaging matrix $\mathbf{M}$ (drawn as *heatmaps*). On the *left*, we show the details for a 32-worker ring topology, and on the *right*, we compare it to more connected topologies. Increasing $\gamma$ (and with it $n_{\mathbf{W}}(\gamma)$) initially leads to larger learning rates thanks to noise reduction. At the optimum, the cost of consensus exceeds the benefit of further reduced noise.

because the goal of collaboration is to *increase* the learning rate. We should therefore pick $\gamma$ such that the first term in Equation (10) dominates. This intuition is summarized in Corollary II, which compares the performance of D-SGD with centralized SGD with fewer workers.

**Corollary II.** D-SGD is as fast as centralized mini-batch SGD with $\mathcal{O}(n_{\mathbf{W}}(\gamma))$ workers, assuming that $\zeta \geq nL$, and that the parameter $\gamma$ is the highest $\gamma$ such that $\frac{2n_{\mathbf{W}}(\gamma)^2}{1-\gamma\lambda_2(\mathbf{W})} \leq 32\frac{\zeta}{L}$. This corresponds to a learning rate $\eta = n_{\mathbf{W}}(\gamma)/16\zeta$.

The typical D-SGD learning rates [8] are of order $\mathcal{O}\big(\min(1/T, 1-\lambda_2(W))\big)$, which are much smaller than the learning rate of Corollary II when $\lambda_2(\mathbf{W})$ is large or the number of iterations large. We use the condition $\zeta \geq nL$ only to present results in a simpler way. The condition $\frac{2n_{\mathbf{W}}(\gamma)^2}{1-\gamma\lambda_2(\mathbf{W})}$ only depends on the size and topology of the graph, and can easily be computed in many cases. Thus, to obtain the best guarantees, we start from $\gamma = 0$ and then increase it until either $n_{\mathbf{W}}(\gamma) \approx n$, the total size of the graph, or the two terms in the minimum match. This is how we obtain Figure 3.

*Proof sketch (Theorem I).* The proof relies on a simple argument: rather than bounding $\|\mathbf{x}^{(t)} - \mathbf{x}^\star\|^2$ or $\|\frac{1}{n}\mathbf{1}\mathbf{1}^\top\mathbf{x}^{(t)} - \mathbf{x}^\star\|^2$, we analyze $\|\mathbf{x}^{(t)} - \mathbf{x}^\star\|_{\mathbf{M}}^2$. This term better captures the benefit of averaging than $\|\mathbf{x}^{(t)} - \mathbf{x}^\star\|^2$, thus leading to better smoothness constants, as long as $\|\mathbf{x}^{(t)}\|_{\mathbf{I}-\mathbf{M}}^2$ is not too large. This yields fast rates without the need to guarantee that iterates between very distant workers remain close, which would be prohibitively expensive. $\qquad\square$

Theorem I is a special case of a more general theorem presented in Appendix D. This version, among other things, covers different choices of parameters, unbalanced communication and computation probabilities (thus allowing for local steps), and the convex ($\mu = 0$) case.

## 5 Experimental analysis

While in the previous sections we have discussed isotropic quadratics or convex and smooth functions, the initial motivation for this work comes from observations in deep learning. First, it is crucial in deep learning to use a large learning rate in the initial phase of training [10]. Contrary to what current theory prescribes, we do not use smaller learning rates in decentralized optimization than when training alone (even when data is heterogeneous.) And second, we find that the spectral gap of a topology is not predictive of the performance of that topology in deep learning experiments.

In this section, we experiment with a variety of 32-worker topologies on Cifar-10 [9] with a VGG-11 model [19]. Like other recent works [13, 22], we opt for this older model, because it does not

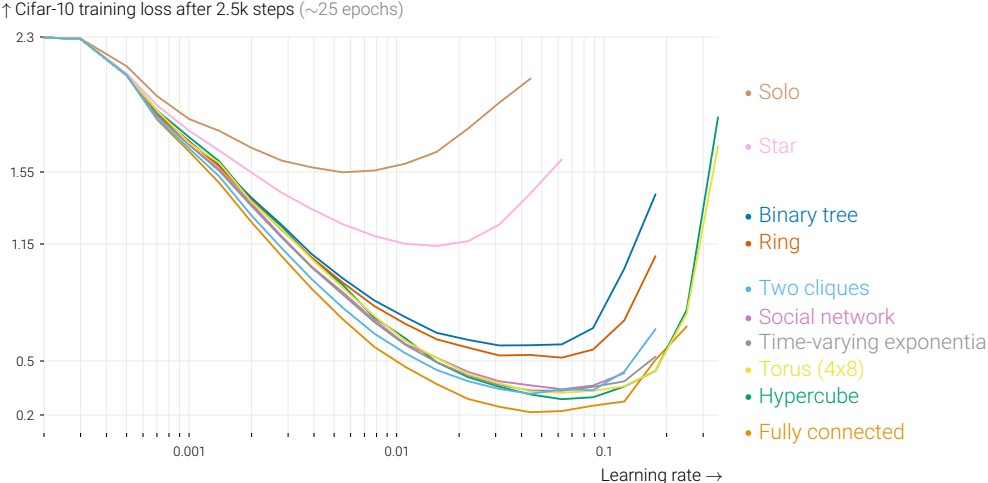

Figure 4: Training loss reached after 2.5k SGD steps with a variety of graph topologies. In all cases, averaging yields a small increase in speed for small learning rates, but a large gain over training alone comes from being able to increase the learning rate. While the star has a better spectral gap (0.031) than the ring (0.013), it performs worse, and does not allow large learning rates. For reference, similar curves for fully-connected graphs of varying sizes are in Appendix F.

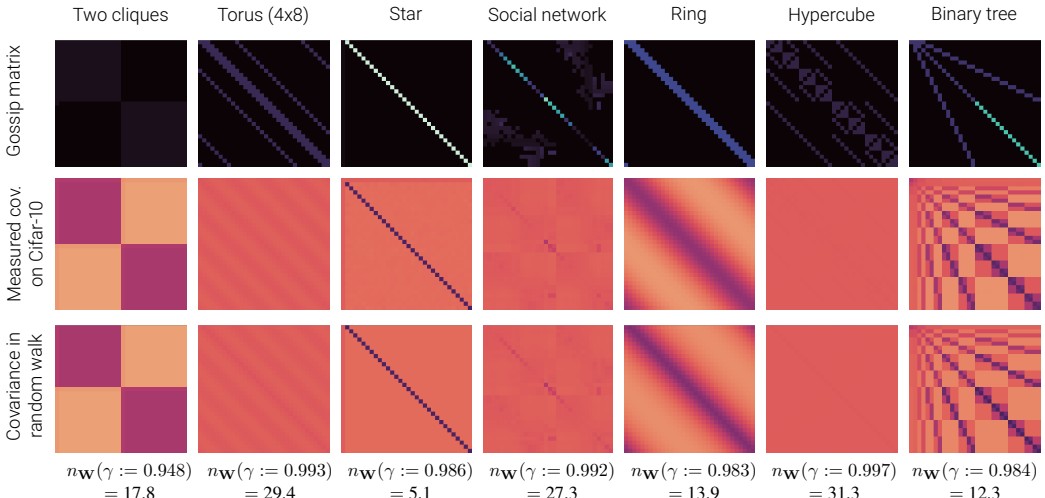

Figure 5: Measured covariance in Cifar-10 (second row) between workers using various graphs (top row). After 10 epochs, we store a checkpoint of the model and train repeatedly for 100 SGD steps, yielding 100 models for 32 workers. We show normalized covariance matrices between the workers. These are very well approximated by the covariance in the random walk process of Section 3 (third row). We print the fitted decay parameters and corresponding 'effective number of neighbors'.

include BatchNorm [6] which forms an orthogonal challenge for decentralized SGD. Please refer to Appendix E for full details on the experimental setup. Our set of topologies (Appendix B) includes regular graphs like rings and toruses, but also irregular graphs such as a binary tree [22] and social network [5], and a time-varying exponential scheme [1]. We focus on the initial phase of training, 25k steps in our case, where both train and test loss converge close to linearly. Using a large learning rate in this phase is found to be important for good generalization [10].

Figure 4 shows the loss reached after the first 2.5k SGD steps for all topologies and for a dense grid of learning rates. The curves have the same global structure as those for isotropic quadratics Figure 1: (sparse) averaging yields a small increase in speed for small learning rates, but a large gain over training alone comes from being able to increase the learning rate. The best schemes support almost the same learning rate as 32 fully-connected workers, and get close in performance.

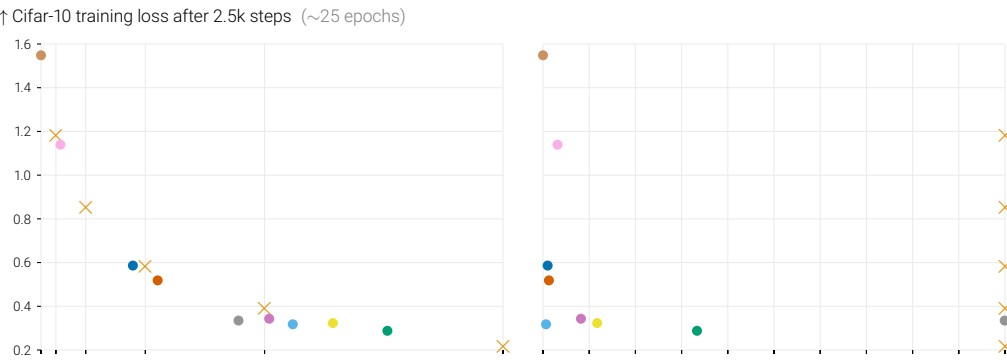

↑ Cifar-10 training loss after 2.5k steps (∼25 epochs)

Effective num. neighbors ($\gamma = 0.951$, tuned) →      Spectral gap →

Figure 6: Cifar-10 training loss after 2.5 k steps for all studied topologies with their optimal learning rates. Colors match Figure 4, and × indicates fully-connected graphs with varying number of workers. After fitting a decay parameter $\gamma = 0.951$ that captures problem specifics, the effective number of neighbors (left) as measured by variance reduction in a random walk (like in Section 3) explains the relative performance of these graphs much better than the spectral gap of these topologies (right).

We also find that the random walks introduced in Section 3 are a good model for variance between workers in deep learning. Figure 5 shows the empirical covariance between the workers after 100 SGD steps. Just like for isotropic quadratics, the covariance is accurately modeled by the covariance in the random walk process for a certain decay rate $\gamma$.

Finally, we observe that the effective number of neighbors computed by the variance reduction in a random walk (Section 3) accurately describes the relative performance under tuned learning rates of graph topologies on our task, including for irregular and time-varying topologies. This is in contrast to the topology's spectral gaps, which we find to be not predictive. We fit a decay rate $\gamma = 0.951$ that seems to capture the specifics of our problem, and show the correlation in Figure 6.

In Appendix F.1, we replicate the same experiments in a different setting. There, we use larger graphs (of 64 workers), a different model and data set (an MLP on Fashion MNIST [24]), and no momentum or weight decay. The results in this setting are qualitatively comparable to the ones presented above.

## 6 Conclusion

We have shown that the sparse averaging in decentralized learning allows larger learning rates to be used, and that it speeds up training. With the optimal large learning rate, the workers' models are not guaranteed to remain close to their global average. Enforcing global consensus is unnecessary in the i.i.d. setting and the small learning rates it would require are counter-productive. With the optimal learning rate, models *do* remain close to some local average in a weighted neighborhood around them. The workers benefit from a number of 'effective neighbors', smaller than the whole graph, that allow them to use a large learning rate while retaining sufficient consensus within the 'local neighborhood'.

Based on our insights, we encourage practitioners of sparse distributed learning to look beyond the spectral gap of graph topologies, and to investigate the actual 'effective number of neighbors' that is used. We also hope that our insights motivate theoreticians to be mindful of assumptions that artificially limit the learning rate.

We show experimentally that our conclusions hold in deep learning, but extending our theory to the non-convex setting is an important open direction that could reveal interesting new phenomena. Furthermore, an extension of our semi-local analysis to the heterogeneous setting where workers optimize different objectives could shed further light on the practical performance of D-SGD.

## Acknowledgments and Disclosure of Funding

This project was supported by SNSF grant 200020_200342.

We thank Lie He for valuable conversations and for identifying the discrepancy between a topology's spectral gap and its empirical performance. We also thank Raphaël Berthier, Aditya Vardhan Varre and Yatin Dandi for their feedback on the manuscript.

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
