- Solo
- Star
- Binary tree
- Ring
- Two cliques
- Social network
- Time-varying exponential
- Torus (4x8)
- Hypercube
- Fully connected

Learning rate →

Figure 4: Training loss reached after 2.5k SGD steps with a variety of graph topologies. In all cases, averaging yields a small increase in speed for small learning rates, but a large gain over training alone comes from being able to increase the learning rate. While the star has a better spectral gap (0.031) than the ring (0.013), it performs worse, and does not allow large learning rates. For reference, similar curves for fully-connected graphs of varying sizes are in Appendix F.

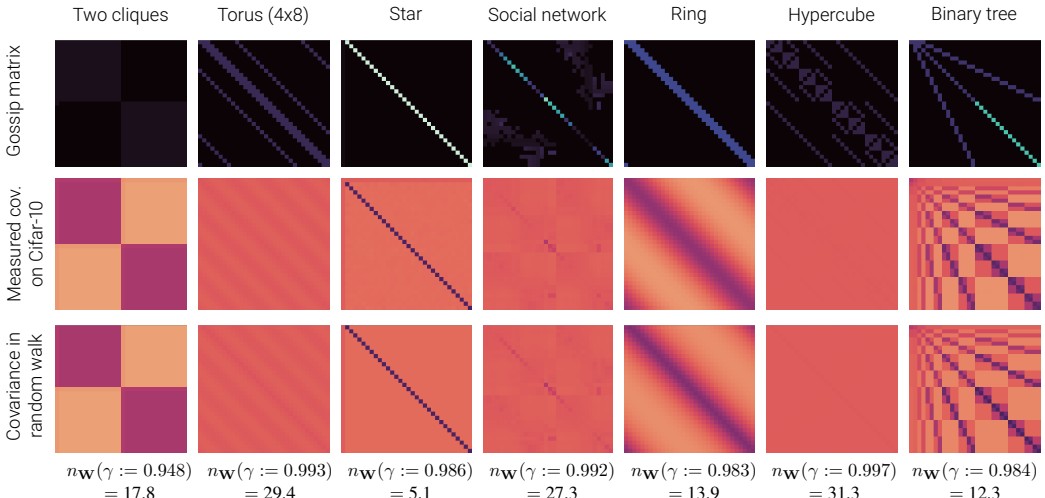

| Two cliques | Torus (4x8) | Star | Social network | Ring | Hypercube | Binary tree |
|---|---|---|---|---|---|---|
| $n_{\mathbf{W}}(\gamma := 0.948)$ $= 17.8$ | $n_{\mathbf{W}}(\gamma := 0.993)$ $= 29.4$ | $n_{\mathbf{W}}(\gamma := 0.986)$ $= 5.1$ | $n_{\mathbf{W}}(\gamma := 0.992)$ $= 27.3$ | $n_{\mathbf{W}}(\gamma := 0.983)$ $= 13.9$ | $n_{\mathbf{W}}(\gamma := 0.997)$ $= 31.3$ | $n_{\mathbf{W}}(\gamma := 0.984)$ $= 12.3$ |

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

## Acknowledgments and Disclosure of Funding

This project was supported by SNSF grant 200020_200342.

We thank Lie He for valuable conversations and for identifying the discrepancy between a topology's spectral gap and its empirical performance. We also thank Raphaël Berthier, Aditya Vardhan Varre and Yatin Dandi for their feedback on the manuscript.

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

# Contents of the Appendix

# A  Notation

Table 1 defines some notation and conventions used throughout this paper and in the appendix.

Table 1: Notation

| | |
|---|---|
| Bold symbol $\mathbf{v}$ | Vector |
| Bold uppercase $\mathbf{M}$ | Matrix |
| $\mathcal{N}^d(0, 1)$ | Standard normal distribution with $d$ independent dimensions |
| $\langle \mathbf{x}, \mathbf{y} \rangle$ | Inner product $\mathbf{x}^\top \mathbf{y}$ |
| $\|\mathbf{T}\|_2$ | Spectral norm |
| $\|\mathbf{T}\|_F$ | Frobenius norm |
| $\mathbf{P} \otimes \mathbf{Q}$ | Kronecker product |
| $\mathbf{1}$ | Vector of all ones |

# B  Topologies

The static topologies that we consider in this work are drawn in Figure 7. Figures 8 and 9 show the gossip matrices we use in detail.

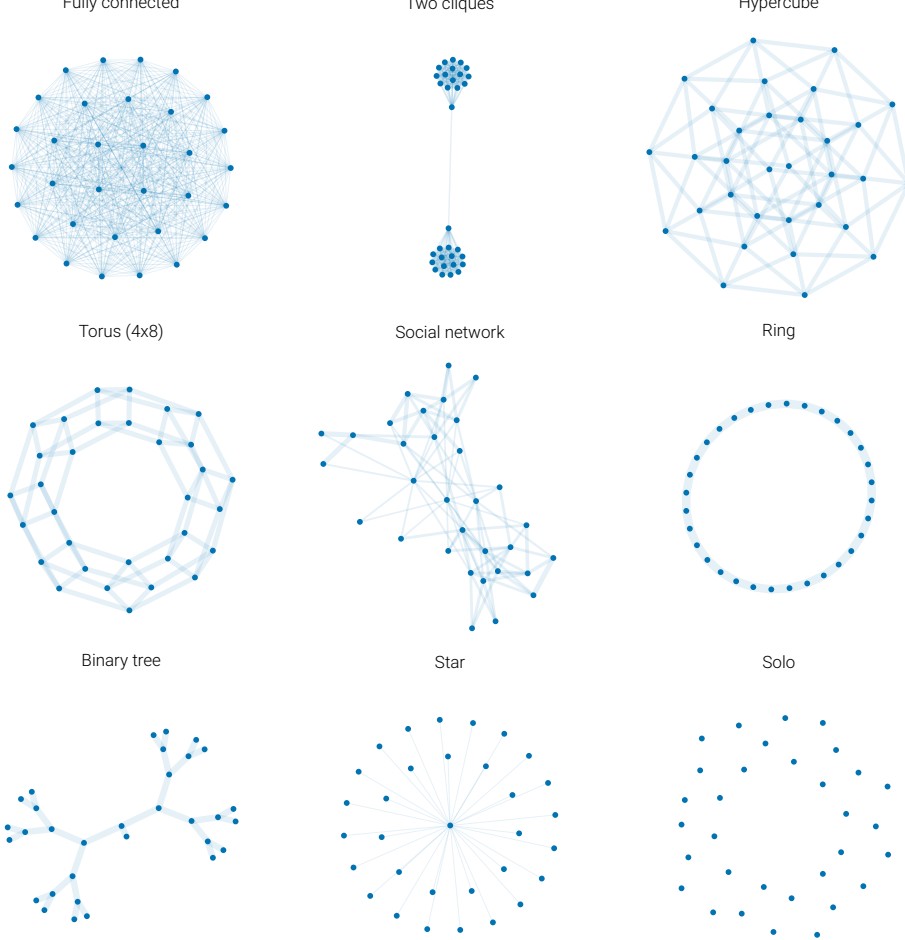

Figure 7: Spring-layout drawings of the static graph topologies considered used this paper. The nodes represent workers, and an edge between two workers indicates that they are connected. The thickness of a edges is proportional to its averaging weight.

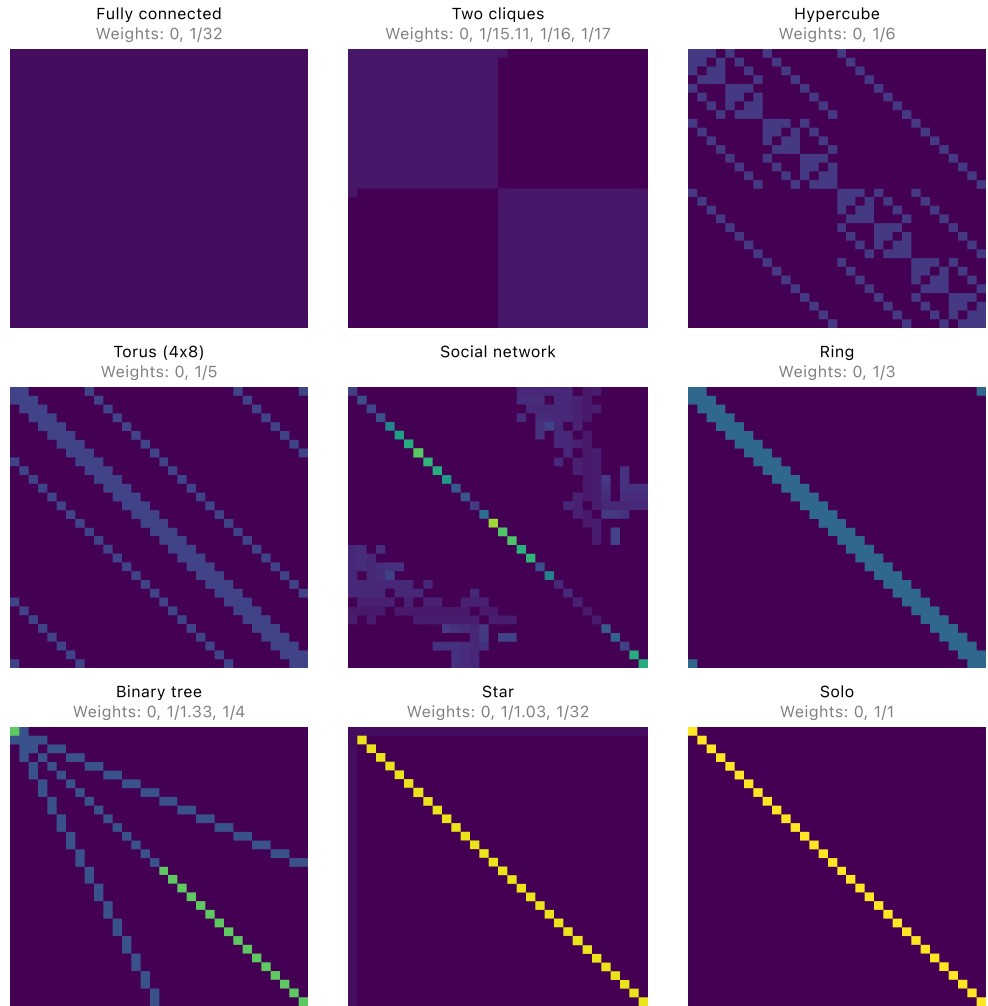

Figure 8: Gossip matrices corresponding to the graph topologies drawn in Figure 7. $x$ and $y$ axes represent workers, and the color of each coordinate in the plots indicates the gossip weight between each pair of workers. The brigher, the higher the weight.



Figure 9: Gossip matrices for the time-varying exponential graph [1, 25]. The product of $\log n$ consecutive gossip matrices equals to the fully-connected averaging matrix with $w_{ij} = 1/n \ \forall i, j$.

## C  Random quadratics

### C.1  Objective

We study the simple problem of minimizing an isotropic $d$-dimensional quadratic,

$$\mathbf{x}^\star = \arg\min_{\mathbf{x} \in \mathbb{R}^d} f(\mathbf{x})$$

where the objective function $f(x) = \frac{1}{2}\|\mathbf{x}\|^2$ is considered to be the expectation over an infinite dataset with random normal features and labels 0:

$$f(\mathbf{x}) = \mathbb{E}_{\mathbf{d} \sim \mathcal{N}^d(0,1)} \frac{1}{2} \langle \mathbf{d}, \mathbf{x} \rangle^2. \tag{12}$$

The optimum of this objective is at $\mathbf{x}^\star = \mathbf{0}$ without loss of generality, because any shifted quadratic would behave the same in the algorithm studied. We will access this objective function through stochastic gradients of the form $\mathbf{g}(\mathbf{x}) = \mathbf{d}\mathbf{d}^\top\mathbf{x}$. The stochasticity of these gradients disappears at the optimum, like in an over-parameterized model.

The difficulty of this problem depends on the dimensionality $d$. For a lower-dimensional problem, the 'stochastic Hessian' $\mathbf{d}\mathbf{d}^\top$ is closer to the true hessian $\mathbf{I}$ than for a high dimensional one. This level of stochasticity is captured by the following quantity:

**Definition C** (Noise level). $\zeta = \sup_{\mathbf{x}} \frac{\mathbb{E}_{\mathbf{d}}\|\mathbf{d}\mathbf{d}^\top\mathbf{x}\|^2}{\|\mathbf{x}\|^2}$.

For our random normal data with batch size 1, this notion of noise level corresponds directly to the dimensionality of the data as $\zeta = d + 2$.

### C.2  Algorithm

The objective (12) is collaboratively optimized by $n$ workers. At every time step $t$, each worker $i$ has its own copy of the 'model' $\mathbf{x}_i^{(t)} \in \mathbb{R}^d$. In the D-SGD algorithm, workers iteratively compute stochastic gradient estimates $\mathbf{g}_i^{(t)} = \mathbf{d}_i^{(t)}\mathbf{d}_i^{(t)\top}\mathbf{x}_i^{(t)}$, where $\mathbf{d}_i^{(t)}$ are i.i.d. from $\mathcal{N}^d(0,1)$. The stochastic gradients are unbiased: $\mathbb{E}\,\mathbf{g}_i^{(t)} = \nabla f(\mathbf{x}_i^{(t)}) = \mathbf{x}_i^{(t)}$.

Workers interleave stochastic gradient updates with gossip averaging:

$$\mathbf{x}_i^{(0)} = \mathbf{x}^{(0)} \quad \forall i$$
$$\mathbf{x}_i^{(t+1)} = \mathbf{W}(\mathbf{x}_i^{(t)} - \eta\mathbf{g}_i^{(t)}),$$

where $\eta$ is the learning rate and

$$\mathbf{W}(\mathbf{x}_i) = \sum_{j=1}^{n} w_{ij}\mathbf{x}_j.$$

This linear operation can be interpreted as matrix multiplication, but one operating on each coordinate of the model independently. $\mathbf{W}$ is an $n \times n$ matrix, and *not* a $d \times d$ matrix as the notation may suggest. The averaging weights $w_{ij}$ encode the connectivity of the communication topology: non-zero $w_{ij}$ implies that workers $i$ and $j$ are directly connected. We make several assumptions about the gossip weights in this analysis:

**Assumption D.**  Constant gossip weights: The weights $w_{ij}$ do not change between steps of D-SGD.

**Assumption E.**  Symmetric gossip weights: $w_{ij} = w_{ji}$.

**Assumption F.**  Doubly stochastic gossip weights: $w_{ij} \geq 0 \,\forall i,j$, $\sum_j w_{ij} = 1 \,\forall i$, $\sum_i w_{ij} = 1 \,\forall j$.

**Assumption G.**  Regular topology: all workers have $k$ directly-connected neighbors, and $w_{ij} = c$ for some constant $c$, and for each edge where $i \neq j$.

**Definition H.**  Spectrum of $\mathbf{W}$. Let the eigenvalues of $\mathbf{W}$ be $\lambda_1 \geq \lambda_2 \geq \ldots \geq \lambda_n$. We call the corresponding eigenvectors $\mathbf{v}_1, \ldots, \mathbf{v}_n$. Under assumption F, $\lambda_1 = 1$, and we call $1 - \lambda_2$ the *spectral gap* of $\mathbf{W}$.

The assumptions on constant gossip weights and regular topologies are mainly here to ease the analysis. We experimentally observe that our findings hold for time-varying topologies and infinite graphs, too, and that they approximately hold for irregular graphs.

## C.3 Linear convergence of an unrolled error vector

We will study the convergence of the algorithm by tracking the *error matrix* $\mathbf{E} \in \mathbb{R}^{n \times n}$. The coordinates of this matrix are the expected covariance between each pair of workers in the network.

$$\mathbf{E}_{ij}^{(t)} = \mathbb{E} \left\langle \mathbf{x}_i^{(t)}, \mathbf{x}_j^{(t)} \right\rangle.$$

We sometimes flatten the error matrix into a vector $\mathbf{e} \in \mathbb{R}^{n^2}$, such that $\mathbf{e}_{ni+j} = \mathbf{E}_{ij}$. The diagonal entries of this matrix describe the worker's error on the objective, and as all workers converge to the optimum at zero, each entry of the matrix will converge to zero. Our analysis of $\mathbf{E}$ quantity starts with a key observation:

**Lemma 1.** There exists an $n^2 \times n^2$ 'transition' matrix $\mathbf{T}$ such that $\mathbf{e}^{(t+1)} = \mathbf{T}\mathbf{e}^{(t)} \ \forall t$.

*Proof.* Because both gossip averaging and the gradient updates are linear, this follows from expanding the inner product. $\qquad\square$

The transition matrix $\mathbf{T}$ depends on the gossip matrix $\mathbf{W}$ and on the learning rate $\eta$. Its spectral gap describes the convergence of the algorithm. D-SGD converges linearly if the norm $\|\mathbf{T}\|_2 < 1$.

We separate $\mathbf{T}$ into a product $\mathbf{T} = \mathbf{T}^{\text{gossip}}\mathbf{T}^{\text{grad}}$, where $\mathbf{T}^{\text{grad}}$ and $\mathbf{T}^{\text{gossip}}$ respectively capture the gradient update and gossip steps of the algorithm. We find that

$$\mathbf{T}^{\text{gossip}} = \mathbf{W} \otimes \mathbf{W}$$

and that $\mathbf{T}^{\text{grad}}$ is diagonal. It only operates element-wise, such that

$$\left[\mathbf{T}^{\text{grad}}\mathbf{e}\right]_{ni+j} = \begin{cases} (1-\eta)^2 \mathbf{e}_{ni+j} + (\zeta - 1)\eta^2 \mathbf{e}_{ni+j} & i = j \text{ (same worker)}, \\ (1-\eta)^2 \mathbf{e}_{ni+j} & i \neq j \text{ (different workers)}. \end{cases} \tag{13}$$

This follows directly from expanding the inner product $\langle \mathbf{x}_i - \eta\mathbf{g}_i, \mathbf{x}_j - \eta\mathbf{g}_j \rangle$. The terms with $i = j$ behave differently than the ones where $i \neq j$, because the noise cancels if $i \neq j$.

## C.4 Random walks with gossip averaging

Before we study the convergence of D-SGD on the random quadratic objective, we first take a step back and inspect a particular random walk process, where workers average their random walk iterates through gossip averaging.

Let $\mathbf{z}^{(t)} \in \mathbb{R}^n$ be a vector containing (scalar) iterates of $n$ workers in the following process:

$$\mathbf{z}^{(0)} = \mathbf{0}, \tag{14}$$

$$\mathbf{z}^{(t+1)} = \mathbf{W}\left(\sqrt{\gamma}\mathbf{z}^{(t)} + \boldsymbol{\xi}^{(t)}\right) \quad \text{where } \boldsymbol{\xi}^{(t)} \sim \mathcal{N}^n(0, 1). \tag{15}$$

We call the parameter $0 < \gamma \leq 1$ the 'decay rate'. Note that the name *random walk* refers to iterative addition of random noise to the workers iterates, and not to a 'random walk' between nodes of the graph.

For this random walk, we will track the covariance matrix $\mathbf{C} \in \mathbb{R}^{n \times n}$ across workers (and its flattened version $\mathbf{c} \in \mathbb{R}^{n^2}$). Its coordinates are

$$\mathbf{C}_{ij}^{(t)} = \mathbb{E}[\mathbf{z}_i^{(t)}\mathbf{z}_j^{(t)}].$$

**Lemma 2.** For static, symmetric and doubly-stochastic topologies (Assumptions D, E and F), the Eigen decomposition of the covariance is

$$\mathbf{C}^{(t)} = \sum_{i=1}^{n} c_i^{(t)} \mathbf{v}_i \mathbf{v}_i^\top,$$

with $0 \leq c_i^{(t)} \leq \frac{\lambda_i^2}{1-\gamma\lambda_i^2}$. Here $(\lambda_i, \mathbf{v}_i)$ are the eigenvalue/eigenvector pairs of $\mathbf{W}$. As $t \to \infty$, $c_i^{(t)} = \frac{\lambda_i^2}{1-\gamma\lambda_i^2}$ with equality.

*Proof.* We can unroll the iterations:

$$\mathbf{z}^{(t)} = \sum_{k=1}^{t} \mathbf{W}^k \gamma^{(k-1)/2} \boldsymbol{\xi}^{(t-k)}$$

and use the temporal independence of $\boldsymbol{\xi}^{(t)}$ to write

$$\mathbf{C}^{(t)} = \mathbb{E}[\mathbf{z}^{(t)}\mathbf{z}^{(t)\top}] = \sum_{k=1}^{t} \gamma^k \mathbf{W}^{2k} \, \mathbb{E}[\boldsymbol{\xi}^{(t-k)}\boldsymbol{\xi}^{(t-k)\top}] = \sum_{k=1}^{t} \gamma^k \mathbf{W}^{2k}.$$

Using commutativity of $\mathbf{W}$ and its Eigen decomposition (Assumptions E, F), we can decompose it as

$$\mathbf{C}^{(t)} = \sum_{i=1}^{n} \mathbf{v}_i \mathbf{v}_i^\top \underbrace{\left( \sum_{k=1}^{t} \gamma^{k-1} \lambda_i^{2k} \right)}_{c_i^{(t)}}$$

Because all terms of parenthesized expression are non-negative, and its limit equals $\frac{\lambda_i^2}{1-\gamma\lambda_i^2}$, this proves the Lemma. $\square$

**Lemma 3.** When the topology is regular (Assumption G) in addition to the assumptions of Lemma 2, workers in the random walk process have equal variance:

$$\mathrm{Var}[\mathbf{z}_i^{(t)}] = \frac{1}{n} \mathrm{Tr}[\mathbf{C}^{(t)}] = \frac{1}{n} \sum_{i=1}^{n} c_i^{(t)}.$$

*Proof.* The variances of $\mathbf{z}_i$ are the diagonal entries of the covariance matrix. By regularity, and because workers are initialized equally, all workers should have the same variance. $\mathrm{Var}[\mathbf{z}_i^{(t)}]$ is therefore equal to the average diagonal entry of $\mathbf{C}_i^{(t)}$. The second equality is a standard property of the trace. $\square$

**Lemma 4.** Under the assumptions of Lemma 3, the variance $\mathrm{Var}[\mathbf{z}_i^{(t)}]$ increases over time:

$$\mathrm{Var}[\mathbf{z}_i^{(t)}] \leq \mathrm{Var}[\mathbf{z}_i^{(t+1)}] \leq \lim_{t' \to \infty} \mathrm{Var}[\mathbf{z}_i^{(t')}] \qquad \forall t.$$

*Proof.* If we write $\mathrm{Var}[\mathbf{z}_i^{(t)}]$ as $\frac{1}{n}\sum_{i=1}^{n} c_i^{(t)}$ using Lemma 3, the statement of this Lemma follows from the realization in Lemma 2 that $c_i^{(t)}$ increases over time to the limit $\frac{\lambda_i^2}{1-\gamma\lambda_i^2}$ for all $i$. $\square$

Note that while the results above are for static gossip matrices, random walks and these variance quantities can be analogously defined time-varying topologies. Those just lack a simple exact form. The stronger the averaging of the gossip process, the lower the variance. We capture this in the following quantity:

**Definition I** (Effective number of neighbors).

$$n_{\mathbf{W}}(\gamma) = \frac{\frac{1}{1-\gamma}}{\lim_{t \to \infty} \frac{1}{n} \sum_{i=1}^{n} \mathrm{Var}[\mathbf{z}_i^{(t)}]},$$

where $\mathbf{z}$ are the iterates from a random walk with gossip averaging, with decay parameter $\gamma$. The numerator is the variance of the random walk process without any gossip averaging ($\mathbf{W} = \mathbf{I}$).

## C.5 Converging random walk

The covariance of the random walk process $\mathbf{C}$ and the error matrix of D-SGD iterates $\mathbf{E}$ share clear similarities. The quantities are both iteratively updated by an affine transformation. The main difference between them, however, is that $\mathbf{C}$ converges to a non-zero constant while $\mathbf{E}$ converges linearly to zero (or it diverges.)

In the next section, we draw a clear connection between the two processes, but first, we define a modified version of the random walk process that further highlights their similarity.

**Definition J** (Scaled random walk). Let $0 < r < 1$ be a scalar. We define a scaled version of the random walk iterates, such that

$$\mathbf{y}^{(t)} = (1 - r)^{t/2}\mathbf{z}^{(t)},$$
$$\mathbf{B}^{(t)} = (1 - r)^t\mathbf{C}^{(t)}, \text{ and}$$
$$\operatorname{Var}[\mathbf{y}_i^{(t)}] = (1 - r)^t \operatorname{Var}[\mathbf{z}_i^{(t)}]$$

Because the sequence $\mathbf{z}^{(t)}$ converges to a non-zero stationary point, the scaled sequence $\mathbf{y}^{(t)}$ converges to zero with a linear rate $r$.

**Lemma 5.** Under the assumptions of Lemma 3, the variance $\operatorname{Var}[\mathbf{y}_i^{(t)}]$ is bounded as

$$\operatorname{Var}[\mathbf{y}_i^{(t)}] \leq \frac{(1 - r)^t}{(1 - \gamma)n\mathbf{w}(\gamma)},$$

with equality as $t \to \infty$.

*Proof.* From Lemma 4, we know that $(1 - r)^t \operatorname{Var}[\mathbf{z}_i^{(t)}] \leq (1 - r)^t \lim_{t' \to \infty} \operatorname{Var}[\mathbf{z}_i^{(t')}]$, with equality as $t \to \infty$. Because the variance $\operatorname{Var}[\mathbf{z}_i^{(t)}]$ is equal across workers $i$ (Lemma 3), the Lemma follows from rearranging Definition I. $\quad\square$

**Lemma 6.** The covariance vector $\mathbf{b}$ (the flattened version of $\mathbf{B}$) of this scaled random walk process follows the recursion $\mathbf{b}^{(t+1)} = \mathbf{T}^{\text{gossip}}u_t(\mathbf{b}^{(t)})$, where

$$u_t(\mathbf{b}^{(t)})_{ni+j} = \begin{cases} \gamma(1 - r)\,\mathbf{b}_{ni+j}^{(t)} + (1 - r)^{t+1} & i = j \text{ (same worker)}, \\ \gamma(1 - r)\,\mathbf{b}_{ni+j}^{(t)} & i \neq j \text{ (different workers)}. \end{cases}$$

*Proof.* The entries $u_t(\mathbf{b}^{(t)})_{ni+j}$ are inner products:

$$u_t(\mathbf{b}^{(t)})_{ni+j} = (1 - r)^{t+1} \left\langle \sqrt{\gamma}\mathbf{y}_i^{(t)} + \boldsymbol{\xi}_i^{(t)}, \sqrt{\gamma}\mathbf{y}_j^{(t)} + \boldsymbol{\xi}_j^{(t)} \right\rangle$$
$$= \gamma(1 - r)\mathbf{b}_{ni+j}^{(t)} + (1 - r)^{t+1} \mathbb{E}\left\langle \boldsymbol{\xi}_i^{(t)}, \boldsymbol{\xi}_j^{(t)} \right\rangle.$$

The inner product between noise contributions $\boldsymbol{\xi}_i^{(t)}$ and $\boldsymbol{\xi}_j^{(t)}$ are 1 if $i = j$ and 0 otherwise. $\quad\square$

**Lemma 7.** The covariance $\mathbf{b}$ follows the recursion $\mathbf{b}^{(t+1)} \geq \mathbf{T}^{\text{gossip}}\mathbf{T}^{\text{r.w.}}\mathbf{b}^{(t)}$ (element-wise), where

$$[\mathbf{T}^{\text{r.w.}}\mathbf{b}^{(t)}]_{ni+j} = \begin{cases} \gamma(1 - r)\,\mathbf{b}_{ni+j}^{(t)} + (1 - r)(1 - \gamma)n\mathbf{w}(\gamma)\mathbf{b}_{ni+j}^{(t)} & i = j, \\ \gamma(1 - r)\,\mathbf{b}_{ni+j}^{(t)} & i \neq j. \end{cases} \tag{16}$$

In the limit of $t \to \infty$, this is true with equality.

*Proof.* From Lemma 5, we have that $\mathbf{b}_{ni+i}^{(t)} = \operatorname{Var}[\mathbf{y}_i^{(t)}] \leq \frac{(1-r)^t}{(1-\gamma)n\mathbf{w}(\gamma)}$, with equality as $t \to \infty$. The entries of $\mathbf{T}^{\text{r.w.}}\mathbf{b}^{(t)}$ are therefore all smaller than or equal to the entries of $u_t(\mathbf{b}^{(t)})$ from Lemma 6, which proves the Lemma. $\quad\square$

### C.6 The rate for D-SGD

**Theorem III** (D-SGD on random quadratics). Under assumptions D, E, F, and G, if the pair of the learning rate $\eta$ and $r$ satisfy

$$r = 1 - (1-\eta)^2 - \frac{(\zeta-1)\eta^2}{n_{\mathbf{W}}\left(\frac{(1-\eta)^2}{1-r}\right)}, \tag{17}$$

the error of D-SGD with learning rate $\eta$ on the random quadratic objective with noise parameter $\zeta$ converges with rate $r$:

$$\sum_{i=1}^{n} \mathbb{E}\|\mathbf{x}_i^{(t)}\|^2 \le (1-r)^t \sum_{i=1}^{n} \mathbb{E}\|\mathbf{x}_i^{(0)}\|^2.$$

This rate becomes exact as $t \to \infty$.

*Proof.* If the condition (17) is satisfied, the expected error iterates $\mathbf{E}$ (Equation 13) of the D-SGD algorithm follow the transition matrix (16) of Lemma 7 with $\gamma = \frac{(1-\eta)^2}{1-r}$. The choice of $\gamma$ ensures that $\gamma(1-r) = (1-\eta)^2$, and the condition (17) that $(\zeta-1)\eta^2 = (1-r)(1-\gamma)n_{\mathbf{W}}(\gamma)$.

From Lemma 7, we know that a sequence that has this transition matrix $\mathbf{T}^{\text{gossip}}\mathbf{T}^{\text{r.w.}}\mathbf{b}^{(t)}$ (16) converges at least as fast as the iterates of the corresponding scaled random walk process $\mathbf{B}$, with equality in the limit as $t \to \infty$.

Since $\mathbf{B}$ converges to zero with a rate $r$, this now implies the same rate for the error matrix $\mathbf{E}$. The sum $\sum_{i=1}^{n} \mathbb{E}\|\mathbf{x}_i^{(t)}\|^2$ in the statement of this theorem is the trace of the matrix $\mathbf{E}^{(t)}$, and therefore it converges to zero with the same rate. This completes the proof. $\square$

# D  (Strongly)-Convex case, missing proofs and additional results

## D.1  Preliminaries on Bregman divergences

Throughout this section, we will use Bregman divergences, which are defined for a differentiable function $h$ and two points $x, y \in \mathbb{R}^d$ as:

$$D_h(x, y) = h(x) - h(y) - \nabla h(y)^\top (x - y). \tag{18}$$

We assume throughout this paper that the functions we consider are twice continuously differentiable and strictly convex on $\text{dom } h$, and that $\nabla h(x) = \min_y h(y) - x^\top y$ is uniquely defined (although milder assumptions could be used). Among the many properties of these divergences, an important one is that if $h$ is $L$ smooth and $\mu$ strongly-convex, then

$$\frac{\mu}{2}\|x - y\|^2 \leq D_h(x, y) \leq \frac{L}{2}\|x - y\|^2 \tag{19}$$

Another important property is called duality, which states that:

$$D_h(x, y) = D_{h^*}(\nabla h(y), \nabla h(x)), \tag{20}$$

where $h^*$ is the convex conjugate of $h$.

## D.2  Main result

This section is devoted to proving Theorem IV, from which Theorem I can be deduced directly by taking $\omega = M_0$ and $p = 1/2$. We recall Assumption B, which is at the heart of Theorem IV.

**Assumption B.** The stochastic gradients are such that: (I) $\xi_i^{(t)}$ and $\xi_j^{(\ell)}$ are independent for all $t, \ell$ and $i \neq j$. (II) $\mathbb{E}\left[f_{\xi_i^{(t)}}\right] = f$ for all $t, i$ (III) $\mathbb{E}\|\nabla f_{\xi_i^{(t)}}(\mathbf{x}^\star)\|^2 \leq \sigma^2$ for all $t, i$, where $\mathbf{x}^\star$ is a minimizer of $f$. (IV) $f_{\xi_i^{(t)}}$ is convex and $\zeta$-smooth for all $t, i$. (V) $f$ is $\mu$-strongly-convex for $\mu \geq 0$ and $L$-smooth.

Note that Assumption B (IV) is stated in this form for simplicity, but it can be relaxed by asking directly that $\mathbb{E}\left[\|\nabla f_{\xi,i}(x^{(t)}) - \nabla f_{\xi,i}(x^\star)\|^2\right] \leq 2\zeta D_f(x^\star, x^{(t)})$, which can also be implied by assuming that each $f_\xi$ is $\zeta_\xi$-smooth, with $\mathbb{E}\left[\zeta_\xi D_{f_\xi}(x^\star, x^{(t)})\right] \leq \zeta D_f(x^\star, x^{(t)})$ (see Equation (28)). These weaker forms would be satisfied by the toy problem of Section 3.

**Theorem IV.** Denote $\mathbf{x}^{(t)}$ the iterates obtained by D-SGD, $\mathbf{L_M} = \mathbf{I} - \mathbf{M}$, and $p$ the probability to perform a communication step ($\mathbf{x}_{t+1} = \mathbf{W}\mathbf{x}_t$). Parameter $\beta$ is such that $\mathbf{I} - \mathbf{W} \succcurlyeq \beta \mathbf{L_M}$. For some $\omega > 0$, denote:

$$\mathcal{L}_t = \|\mathbf{x}^{(t)} - \mathbf{x}^\star\|_{\mathbf{M}}^2 + \omega\|\mathbf{x}^{(t)}\|_{\mathbf{L_M}}^2. \tag{21}$$

Then, if $\eta$ is such that:

$$\eta \leq \frac{M_0 \beta}{L}\frac{p}{1-p}, \tag{22}$$

$$\eta \leq \frac{1}{4(M_0\zeta + L)} \tag{23}$$

we have that:

$$\mathcal{L}_t \leq [1 - (1-p)\eta\mu]^t \mathcal{L}_0 + \frac{\eta\tilde{\sigma}^2}{\mu}, \tag{24}$$

with $\tilde{\sigma}^2 = \sigma_{\mathbf{M}}^2 + \omega\sigma_{\mathbf{L_M}}^2$, where $\mathbb{E}\|\nabla f_{\xi_i^{(t)}}(\mathbf{x}^\star)\|_{\mathbf{M}}^2 \leq \sigma_{\mathbf{M}}^2$ (and similarly for $\mathbf{L_M}$).

In the convex case ($\mu = 0$), we have:

$$\mathbb{E}\left[\frac{1}{T}\sum_{t=0}^{T-1} D_f(\mathbf{M}\mathbf{x}^{(t)}, \mathbf{x}^\star)\right] \leq \frac{1}{1-p}\frac{\mathcal{L}_0}{\eta T} + \eta\tilde{\sigma}^2 \tag{25}$$

Note that the factors 2 in Equation (23) are simplifications to make the result more readable but could be improved.

*Proof.* We now proceed to the proof of the theorem. To show that the Lyapunov $\mathcal{L}_t$ decreases over iterations, we will study how each quantity $\|\mathbf{x}^{(t)} - \mathbf{x}^\star\|_{\mathbf{M}}^2$ and $\|\mathbf{x}^{(t)}\|_{\mathbf{L_M}}^2$ evolves through time. In particular, we will first consider the case of computation updates (so, local gradient updates), and then the case of gossip updates.

**1 - Computation updates**   In this case, we assume that the update is of the form

$$\mathbf{x}^{(t+1)} = \mathbf{x}^{(t)} - \eta \nabla f_{\xi_t}(\mathbf{x}^{(t)}). \tag{26}$$

This happens with probability $1 - p$, and expectations are taken with respect to $\xi_t$. To avoid notation clutter, we use notations $\nabla f_\xi$ and $\nabla f_{\xi,i}$, which are such that $\nabla f_{\xi,i}(\mathbf{x}^{(t)}) = (\nabla f_\xi(\mathbf{x}^{(t)}))_i = \nabla f_{\xi_i^{(t)}}(\mathbf{x}_i^{(t)})$.

**Distance to optimum**   We bound the distance to optimum as follows, using that $\mathbf{Mx}^\star = \mathbf{x}^\star$, and $\mathbb{E}\left[\nabla f_\xi(\mathbf{x}^{(t)})\right] = \nabla f(\mathbf{x}^{(t)})$:

$$\mathbb{E}\left[\|\mathbf{x}^{(t+1)} - \mathbf{x}^\star\|_{\mathbf{M}}^2\right] = \|\mathbf{x}^{(t)} - \mathbf{x}^\star\|_{\mathbf{M}}^2 - 2\eta\, \mathbb{E}\left[(\mathbf{x}^{(t)} - \mathbf{x}^\star)^\top \mathbf{M}\nabla f_\xi(\mathbf{x}^{(t)})\right] + \eta^2 \|\nabla f_\xi(\mathbf{x}^{(t)})\|_{\mathbf{M}}^2$$

$$= \|\mathbf{x}^{(t)} - \mathbf{x}^\star\|_{\mathbf{M}}^2 - 2\eta(\mathbf{Mx}^{(t)} - \mathbf{x}^\star)^\top \nabla f(\mathbf{x}^{(t)}) + \eta^2\, \mathbb{E}\left[\|\nabla f_\xi(\mathbf{x}^{(t)})\|_{\mathbf{M}}^2\right].$$

Then, we expand the middle term in the following way:

$$-\nabla f(\mathbf{x}^{(t)})^\top (\mathbf{Mx}^{(t)} - \mathbf{x}^\star) = -\nabla f(\mathbf{x}^{(t)})^\top (\mathbf{x}^{(t)} - \mathbf{x}^\star) - \nabla f(\mathbf{x}^{(t)})^\top (\mathbf{Mx}^{(t)} - \mathbf{x}^{(t)})$$

$$= -D_f(\mathbf{x}^{(t)}, \mathbf{x}^\star) - D_f(\mathbf{x}^\star, \mathbf{x}^{(t)}) + D_f(\mathbf{Mx}^{(t)}, \mathbf{x}^{(t)}) - f(\mathbf{Mx}^{(t)}) + f(\mathbf{x}^{(t)})$$

$$= -D_f(\mathbf{Mx}^{(t)}, \mathbf{x}^\star) - D_f(\mathbf{x}^\star, \mathbf{x}^{(t)}) + D_f(\mathbf{Mx}^{(t)}, \mathbf{x}^{(t)})$$

$$\leq -\frac{\mu}{2}\|\mathbf{x}^{(t)} - \mathbf{x}^\star\|_{\mathbf{M}^2}^2 - D_f(\mathbf{x}^\star, \mathbf{x}^{(t)}) + \frac{L}{2}\|\mathbf{Mx}^{(t)} - \mathbf{x}^{(t)}\|^2, \tag{27}$$

where in the last time we used the $\mu$-strong convexity and $L$-smoothness of $f$. For the noise term, we use that fact that $(\nabla f_\xi(\mathbf{x}^{(t)}))_i$ and $(\nabla f_\xi(\mathbf{x}^{(t)}))_j$ are independent for $i \neq j$, so that

$$\frac{1}{2}\, \mathbb{E}\left[\|\nabla f_\xi(\mathbf{x}^{(t)})\|_{\mathbf{M}}^2\right] = \mathbb{E}\left[\|\nabla f_\xi(\mathbf{x}^{(t)}) - \nabla f_\xi(\mathbf{x}^\star)\|_{\mathbf{M}}^2\right] + \mathbb{E}\left[\|\nabla f_\xi(\mathbf{x}^\star)\|_{\mathbf{M}}^2\right]$$

$$= \sum_{i=1}^{n} \mathbf{M}_{ii}\, \mathbb{E}\left[\|\nabla f_{\xi,i}(\mathbf{x}^{(t)}) - \nabla f_{\xi,i}(\mathbf{x}^\star)\|^2\right] + \mathbb{E}\left[\|\nabla f_\xi(\mathbf{x}^\star)\|_{\mathbf{M}}^2\right]$$

$$+ \sum_{i=1}^{n} \sum_{j \neq i} \mathbf{M}_{ij}\, \mathbb{E}\left[[\nabla f_{\xi,i}(\mathbf{x}^{(t)}) - \nabla f_{\xi,i}(\mathbf{x}^\star)]^\top [\nabla f_{\xi,j}(\mathbf{x}^{(t)}) - \nabla f_{\xi,j}(\mathbf{x}^\star)]\right]$$

$$= \sum_{i=1}^{n} \mathbf{M}_{ii}\, \mathbb{E}\left[\|\nabla f_{\xi,i}(\mathbf{x}^{(t)}) - \nabla f_{\xi,i}(\mathbf{x}^\star)\|^2\right] + \|\nabla f(\mathbf{x}^{(t)})\|_{\mathbf{M}}^2 + \mathbb{E}\left[\|\nabla f_\xi(\mathbf{x}^\star)\|_{\mathbf{M}}^2\right].$$

We now use for all $i \in \{1, \ldots, n\}$ the $\zeta$-smoothness of $f_{\xi,i}$, which implies the $\zeta^{-1}$-strong convexity of $f_{\xi,i}^*$ [7], so that:

$$\mathbb{E}\left[\|\nabla f_{\xi,i}(\mathbf{x}^{(t)}) - \nabla f_{\xi,i}(\mathbf{x}^\star)\|^2\right] = 2\, \mathbb{E}\left[D_{\frac{1}{2}\|\cdot\|^2}(\nabla f_{\xi,i}(\mathbf{x}^{(t)}), \nabla f_{\xi,i}(\mathbf{x}^\star))\right]$$

$$\leq \frac{2}{\zeta^{-1}}\, \mathbb{E}\left[D_{f_{\xi,i}^*}(\nabla f_{\xi,i}(\mathbf{x}^{(t)}), \nabla f_{\xi,i}(\mathbf{x}^\star))\right]$$

$$\leq 2\zeta\, \mathbb{E}\left[D_{f_{\xi,i}}(\mathbf{x}^\star, \mathbf{x}^{(t)})\right] \tag{28}$$

$$= 2\zeta D_f(\mathbf{x}^\star, \mathbf{x}^{(t)})$$

For the expected gradient term, we can use that:

$$\|\nabla f(\mathbf{x}^{(t)})\|_{\mathbf{M}}^2 \leq \|\nabla f(\mathbf{x}^{(t)}) - \nabla f(\mathbf{x}^\star)\|^2 \leq 2L D_f(\mathbf{x}^\star, \mathbf{x}^{(t)}),$$

so that in the end,

$$\mathbb{E}\left[\|\nabla f_\xi(\mathbf{x}^{(t)})\|_{\mathbf{M}}^2\right] \leq 4(\zeta M_0 + L) D_f(\mathbf{x}^\star, \mathbf{x}^{(t)}) + 2\sigma_{\mathbf{M}}^2, \tag{29}$$

where $M_0 = \max_i \mathbf{M}_{ii}$, and $\mathbb{E}\left[\|\nabla f_\xi(\mathbf{x}^\star)\|_{\mathbf{M}}^2\right] \leq \sigma_{\mathbf{M}}^2$, the locally averaged variance at optimum. Plugging this into the main equation, we obtain that:

$$\mathbb{E}\left[\|\mathbf{x}^{(t+1)} - \mathbf{x}^\star\|_{\mathbf{M}}^2\right] \leq \|\mathbf{x}^{(t)} - \mathbf{x}^\star\|_{\mathbf{M}}^2 - \eta\mu\|\mathbf{x}^{(t)} - \mathbf{x}^\star\|_{\mathbf{M}^2}^2 + 2\eta^2\sigma_{\mathbf{M}}^2$$

$$- 2\eta\left(1 - 2\eta\left[\zeta M_0 + L\right]\right) D_f(\mathbf{x}^\star, \mathbf{x}^{(t)}) + \frac{L}{2}\|\mathbf{Mx}^{(t)} - \mathbf{x}^{(t)}\|^2$$

The last step is to write that $\mathbf{M}^2 = \mathbf{M} - \mathbf{ML_M}$, so that:

$$\mathbb{E}\left[\|\mathbf{x}^{(t+1)} - \mathbf{x}^\star\|_{\mathbf{M}}^2\right] \leq (1 - \eta\mu)\|\mathbf{x}^{(t)} - \mathbf{x}^\star\|_{\mathbf{M}}^2 + \eta\mu\|\mathbf{x}^{(t)} - \mathbf{x}^\star\|_{\mathbf{ML_M}}^2 + 2\eta^2\sigma_{\mathbf{M}}^2$$
$$- 2\eta\left(1 - 2\eta\left[\zeta M_0 + L\right]\right) D_f(\mathbf{x}^\star, \mathbf{x}^{(t)}) + \eta L\|\mathbf{Mx}^{(t)} - \mathbf{x}^{(t)}\|^2$$

At this point, we can use that $\mathbf{ML_M} \leq \mathbf{I}/4$,

$$\mu\|\mathbf{x}^{(t)} - \mathbf{x}^\star\|_{\mathbf{ML_M}}^2 \leq \frac{\mu}{4}\|\mathbf{x}^{(t)} - \mathbf{x}^\star\|^2 \leq \frac{1}{2}D_f(\mathbf{x}^\star, \mathbf{x}^{(t)}), \tag{30}$$

so that

$$\mathbb{E}\left[\|\mathbf{x}^{(t+1)} - \mathbf{x}^\star\|_{\mathbf{M}}^2\right] \leq (1 - \eta\mu)\|\mathbf{x}^{(t)} - \mathbf{x}^\star\|_{\mathbf{M}}^2 + \eta L\|\mathbf{Mx}^{(t)} - \mathbf{x}^{(t)}\|^2$$
$$- 2\eta\left(3/4 - 2\eta\left[\zeta M_0 + L\right]\right) D_f(\mathbf{x}^\star, \mathbf{x}^{(t)}) + 2\eta^2\sigma_{\mathbf{M}}^2 \tag{31}$$

**Distance to consensus**  We now bound the distance to consensus in the case of a communication update. More specifically, we write that:

$$\mathbb{E}\left[\|\mathbf{x}^{(t+1)}\|_{\mathbf{L_M}}^2\right] = \|\mathbf{x}^{(t)}\|_{\mathbf{L_M}}^2 - 2\eta\nabla f(\mathbf{x}^{(t)})^\top \mathbf{L_M}\mathbf{x}^{(t)} + \eta^2\,\mathbb{E}\left[\|\nabla f_{\xi_t}(\mathbf{x}^{(t)})\|_{\mathbf{L_M}}^2\right]$$

Then, we develop the middle term as:

$$-\nabla f(\mathbf{x}^{(t)})^\top \mathbf{L_M}\mathbf{x}^{(t)} = -\nabla f(\mathbf{x}^{(t)})^\top(\mathbf{I} - \mathbf{M})\mathbf{x}^{(t)}$$
$$= \nabla f(\mathbf{x}^{(t)})^\top(\mathbf{Mx}^{(t)} - \mathbf{x}^{(t)})$$
$$= -D_f(\mathbf{Mx}^{(t)}, \mathbf{x}^{(t)}) + f(\mathbf{Mx}^{(t)}) - f(\mathbf{x}^{(t)})$$
$$\leq -\frac{\mu}{2}\|\mathbf{Mx}^{(t)} - \mathbf{x}^{(t)}\|^2 + f(\mathbf{Mx}^{(t)}) - f(\mathbf{x}^{(t)})$$

By convexity of $f$ (since the expected function is the same for all workers), we have that

$$f(\mathbf{Mx}^{(t)}) \leq f(\mathbf{x}^{(t)}). \tag{32}$$

We finally decompose $\mathbf{L_M}^2 = \mathbf{L_M}(\mathbf{I} - \mathbf{M})$, so that:

$$-2\eta\nabla f(\mathbf{x}^{(t)})^\top \mathbf{L_M}\mathbf{x}^{(t)} \leq -\eta\mu\|\mathbf{x}^{(t)} - \mathbf{x}^\star\|_{\mathbf{L_M}}^2 + \eta\mu\|\mathbf{x}^{(t)}\|_{\mathbf{ML_M}}^2$$

For the noise term, we obtain exactly the same derivations as in the previous setting, but this time with matrix $\mathbf{L_M} = \mathbf{I} - \mathbf{M}$ instead. Using the same bounding, and $M_{\min} = \min_i(\mathbf{M})_{ii}$, we thus obtain:

$$\mathbb{E}\left[\|\nabla f_\xi(\mathbf{x}^{(t)})\|_{\mathbf{L_M}}^2\right] \leq 4(\zeta(1 - M_{\min}) + L)D_f(\mathbf{x}^\star, \mathbf{x}^{(t)}) + 2\sigma_{\mathbf{L_M}}^2. \tag{33}$$

In particular, we have that:

$$\mathbb{E}\left[\|\mathbf{x}^{(t+1)} - \mathbf{x}^\star\|_{\mathbf{L_M}}^2\right] \leq (1 - \eta\mu)\|\mathbf{x}^{(t)} - \mathbf{x}^\star\|_{\mathbf{L_M}}^2 + \eta\mu\|\mathbf{x}^{(t)}\|_{\mathbf{ML_M}}^2 + 2\eta^2\sigma_{\mathbf{L_M}}^2$$
$$+ 4\eta^2(\zeta(1 - M_{\min}) + L)D_f(\mathbf{x}^\star, \mathbf{x}^{(t)}).$$

Similarly to before, we use that

$$\mu\|\mathbf{x}^{(t)}\|_{\mathbf{ML_M}}^2 = \mu\|\mathbf{x}^{(t)} - \mathbf{x}^\star\|_{\mathbf{ML_M}}^2 \leq \frac{\mu}{4}\|\mathbf{x}^{(t)} - \mathbf{x}^\star\|^2 \leq \frac{1}{2}D_f(\mathbf{x}^\star, \mathbf{x}^{(t)}), \tag{34}$$

so that for computation updates, the distance to consensus evolves as:

$$\mathbb{E}\left[\|\mathbf{x}^{(t+1)} - \mathbf{x}^\star\|_{\mathbf{L_M}}^2\right] \leq (1 - \eta\mu)\|\mathbf{x}^{(t)} - \mathbf{x}^\star\|_{\mathbf{L_M}}^2 + 2\eta\left[\frac{1}{4} + 2\eta(\zeta(1 - M_{\min}) + L)\right]D_f(\mathbf{x}^\star, \mathbf{x}^{(t)}) + 2\eta^2\sigma_{\mathbf{L_M}}^2 \tag{35}$$

Combining Equation (35) with Equation (31) leads to:

$$\mathcal{L}^{(t+1)} \leq (1 - \eta\mu)\mathcal{L}_t + \eta L\|\mathbf{Mx}^{(t)} - \mathbf{x}^{(t)}\|^2 + 2\eta^2\tilde{\sigma}^2$$
$$- \eta\left(1 - 4\eta\left[\zeta(M_0 + \omega(1 - M_{\min})) + (1 + \omega)L\right]\right) D_f(\mathbf{x}^\star, \mathbf{x}^{(t)}), \tag{36}$$

with $\tilde{\sigma}^2 = \sigma_{\mathbf{M}}^2 + \omega\sigma_{\mathbf{L_M}}^2$.

**2 - Communication updates** We write:

$$\|\mathbf{x}^{(t+1)} - \mathbf{x}^\star\|^2_{\mathbf{L_M}} = \|\mathbf{x}^{(t)} - \mathbf{x}^\star\|^2_{\mathbf{W L_M W}}$$

$$\leq \|\mathbf{x}^{(t)} - \mathbf{x}^\star\|^2_{\mathbf{W L_M}}$$

$$= \|\mathbf{x}^{(t)} - \mathbf{x}^\star\|^2_{\mathbf{L_M}} - \|\mathbf{x}^{(t)} - \mathbf{x}^\star\|^2_{\mathbf{L_W L_M}}$$

For distance to optimum part in communication update, we obtain:

$$\|\mathbf{x}^{(t+1)} - \mathbf{x}^\star\|^2_{\mathbf{M}} = \|\mathbf{x}^{(t)} - \mathbf{x}^\star\|^2_{\mathbf{WMW}} \leq \|\mathbf{x}^{(t)} - \mathbf{x}^\star\|^2_{\mathbf{M}} \tag{37}$$

We now introduce $\beta$, the strong convexity of $\mathbf{L_W} = \mathbf{I} - \mathbf{W}$ relative to $\mathbf{L_M}$:

$$\mathbf{L_W} \geq \beta \mathbf{L_M}. \tag{38}$$

Therefore, we obtain that for communication updates,

$$\mathcal{L}^{(t+1)} \leq \mathcal{L}_t - \omega\beta\|\mathbf{x}^{(t)} - \mathbf{x}^\star\|^2_{\mathbf{L_M}^2}. \tag{39}$$

**Putting terms back together** We now put everything together, assuming that communication steps happen with probability $p$ (and so computations steps with probability $1 - p$). Thus, we mix Equations (36) and (39) to obtain:

$$\mathbb{E}\left[\mathcal{L}^{(t+1)}\right] \leq (1 - (1-p)\eta\mu)\mathcal{L}_t + 2(1-p)\eta^2\tilde{\sigma}^2$$

$$+ \left[(1-p)\eta L - \omega p\beta\right]\|\mathbf{x}^{(t)}\|^2_{\mathbf{L_M}^2}$$

$$- \eta(1-p)\left(1 - 4\eta\left[\zeta(M_0 + \omega(1 - M_{\min})) + (1+\omega)L\right]\right)D_f(\mathbf{x}^\star, \mathbf{x}^{(t)}).$$

In particular, we obtain the linear decrease of the Lyapunov $\mathcal{L}_t$ under the following conditions:

$$\eta \leq \frac{\omega\beta}{L}\frac{p}{1-p}$$

$$\eta \leq \frac{1}{4\left(\zeta\left[M_0 + \omega(1 - M_{\min})\right] + (1+\omega)L\right)}$$

Under these conditions, we have that

$$\mathbb{E}\left[\mathcal{L}^{(t+1)}\right] \leq (1 - (1-p)\eta\mu)\mathcal{L}_t + (1-p)\eta^2\tilde{\sigma}^2, \tag{40}$$

and we can simply chain this relation to finish the proof of the theorem. $\qquad\square$

**Convex case** In the convex case ($\mu = 0$), the proof is very similar, except that we keep the $D_f(\mathbf{Mx}^{(t)}, \mathbf{x}^\star)$ term from Equation (27). In particular, under the same step-size conditions as the strongly convex case, this leads to:

$$\mathbb{E}\left[\mathcal{L}^{(t+1)}\right] \leq \mathcal{L}_t + 2(1-p)\eta^2\tilde{\sigma}^2 - \eta(1-p)D_f(\mathbf{Mx}^{(t)}, \mathbf{x}^\star). \tag{41}$$

This leads to:

$$\mathbb{E}\left[\frac{1}{T}\sum_{t=0}^{T-1}D_f(M\mathbf{x}^{(t)}, \mathbf{x}^\star)\right] \leq \frac{1}{1-p}\frac{\mathcal{L}_0}{\eta T} + 2\eta\tilde{\sigma}^2, \tag{42}$$

which finishes the proof of the theorem.

**Evaluating $\beta$** There are two important graph quantities: $M_0$ and $\beta$. If we choose $M$ as in Equation (9), then its eigenvalues are equal to $\frac{(1-\gamma)\lambda_i^2}{1-\gamma\lambda_i^2}$, where $\lambda_i$ is the $i$-th eigenvalue of $\mathbf{W}$. Therefore,

$$\lambda_i^{\mathbf{L_M}} = \frac{1 - \lambda_i^2}{1 - \gamma\lambda_i^2}. \tag{43}$$

In particular, we have that for all $i$,

$$1 - \lambda_i \geq \beta\frac{1 - \lambda_i^2}{1 - \gamma\lambda_i^2}, \tag{44}$$

so that we can take

$$\beta = \frac{1 - \gamma\lambda_2^2}{1 + \lambda_2} \geq \frac{1 - \gamma\lambda_2}{2}, \tag{45}$$

where we use $\lambda_2 \leq 1$ to simplify the results. In particular, $\beta$ does not depend on the spectral gap of $\mathbf{W}$ (which is equal to $1 - \lambda_2$) as long as $\gamma$ is not too large. Yet, an interesting phenomenon happens: *a larger graph also implies more effective neighbors for a given $\gamma$.*

**Choice of** $\omega$ A reasonable value for $\omega$ is to simply take it as $\omega = M_0$. Indeed,

- The second condition almost does not benefit from $\omega \leq M_0$ (factor 2 at most).
- If the first condition dominates, such that taking $\omega \geq M_0$ would loosen it, then instead one can reduce $\gamma$. This will lead to a higher value for both $M_0$ (and so for $\omega$) and $\beta$. Note that, again, increasing $M_0$ does not make the second condition stronger than what it would have been with just increasing $\omega$ by more than a factor 2.

With this choice, we thus obtain that:
$$\eta \leq \min \left( \frac{M_0 \beta}{L} \frac{p}{1-p}, \frac{1}{4 \left( M_0 \zeta (2 - M_{\min}) + (1 + M_0) L \right)} \right), \tag{46}$$
and Theorem IV is obtained by taking $M_0 \leq 1$ and $M_{\min} \geq 0$.

## D.3 Obtaining Corollary II

In this section, we discuss the derivations leading to Corollary II. To do so, we start by making the simplifying assumption that
$$\frac{\zeta}{n} \geq L. \tag{47}$$
Using this, and writing $n_{\mathbf{W}}(\gamma) = 1/M_0$, the condition from Equation (23) simplifies to:
$$\eta \leq \frac{L n_{\mathbf{W}}(\gamma)}{16 \zeta}. \tag{48}$$
We always want this condition to be tight, and not Equation (22) the communication one, which is only there to allow us to use larger values of $n_{\mathbf{W}}(\gamma)$. In particular, we want that:
$$\frac{L n_{\mathbf{W}}(\gamma)}{16 \zeta} \leq \frac{\beta}{n_{\mathbf{W}}(\gamma) L}. \tag{49}$$
When we increase $\gamma$, $n_{\mathbf{W}}(\gamma)$ increases and $\beta$ decreases. We thus want to take the highest $\gamma$ such that (49) is verified (potentially with an equality if $n_{\mathbf{W}}(\gamma) < n$).

## D.4 Deterministic algorithm

So far, we have analyzed the randomized variant of D-SGD, in which at each step, there is a coin flip to decide whether to perform a communication or computation step. We now show how to extend the analysis to the case in which:
$$\mathbf{x}^{(t+1)} = \mathbf{W}\mathbf{x}^{(t)} - \eta \nabla f_\xi(\mathbf{W}\mathbf{x}^{(t)}) \tag{50}$$
Note that D-SGD is often presented as $\mathbf{x}_{t+1} = \mathbf{W}(\mathbf{x}_t - \eta \nabla f_\xi(\mathbf{x}_t))$, but it turns out that the analysis is easier when considering it in the form of Equation (50). Yet, it comes down to the same algorithm (alternating communication and computation steps), and the difference simply is whether the error is evaluated after a communication step or a local gradient step. The results in the previous section did not depend on the value of $\mathbf{x}_t$, so we can perform the same derivations with $\mathbf{W}\mathbf{x}_t$ instead of $\mathbf{x}_t$, so that Equation (36) now writes:
$$\begin{aligned} \mathcal{L}(\mathbf{x}^{(t+1)}) = {} & (1 - \eta \mu)\mathcal{L}(\mathbf{W}\mathbf{x}^{(t)}) + \eta L \|\mathbf{M}\mathbf{W}\mathbf{x}^{(t)} - \mathbf{W}\mathbf{x}^{(t)}\|^2 + 2\eta^2 \tilde{\sigma}^2 \\ & - \eta \left( 1 - 4\eta \left[ \zeta(M_0 + \omega(1 - M_{\min})) + (1 + \omega)L \right] \right) D_f(\mathbf{x}^\star, \mathbf{W}\mathbf{x}^{(t)}), \end{aligned} \tag{51}$$
where $\mathcal{L}(\mathbf{x}) = \|\mathbf{x} - \mathbf{x}^\star\|_{\mathbf{M}}^2 + \omega \|\mathbf{x}\|_{\mathbf{L_M}}^2$, so that $\mathcal{L}^{(t)} = \mathcal{L}(\mathbf{x}^{(t)})$. In particular, choosing $\eta$ such that the second line is always negative (as before) leads to:
$$\mathcal{L}(\mathbf{x}^{(t+1)}) = (1 - \eta \mu)\mathcal{L}(\mathbf{W}\mathbf{x}^{(t)}) + \eta L \|\mathbf{x}^{(t)}\|_{\mathbf{W}\mathbf{L_M}^2 \mathbf{W}}^2 + 2\eta^2 \tilde{\sigma}^2. \tag{52}$$
Similarly, using Equation (39), we obtain that
$$\mathcal{L}(\mathbf{W}\mathbf{x}^{(t)}) \leq \mathcal{L}(\mathbf{x}^{(t)}) - \omega \beta \|\mathbf{x}^{(t)} - \mathbf{x}^\star\|_{\mathbf{L_M}^2}^2. \tag{53}$$
Combining Equations (52) and (53) and using that $\mathbf{W}\mathbf{L_M}^2\mathbf{W} \preccurlyeq \mathbf{L_M}^2$, we obtain:
$$\mathcal{L}^{(t+1)} \leq (1 - \eta \mu)\mathcal{L}^{(t)} + (\eta L - (1 - \eta \mu)\omega \beta)\|\mathbf{x}^{(t)}\|_{\mathbf{L_M}^2}^2 + 2\eta^2 \tilde{\sigma}^2. \tag{54}$$
Thus, we obtain similar guarantees (up to a factor $1 - \eta \mu$ which is small) for the deterministic and randomized algorithms. Note that in this case, constant $\beta$ can be replaced by a slightly better constant $\tilde{\beta}$ which would be such that:
$$\mathbf{L_M}\mathbf{L_W} \geq \tilde{\beta} \, \mathbf{W}\mathbf{L_M}^2\mathbf{W}. \tag{55}$$

# E   Cifar-10 experimental setup

Table 2 describes the details of our experiments with D-SGD with VGG-11 on Cifar-10.

Table 2:  Default experimental settings for Cifar-10/VGG-11

| | |
|---|---|
| Dataset | Cifar-10 [9] |
| Data augmentation | Random horizontal flip and random $32 \times 32$ cropping |
| Data normalization | Subtract mean $(0.4914, 0.4822, 0.4465)$ and divide standard deviation $(0.2023, 0.1994, 0.2010)$ |
| Architecture | VGG-11 [19] |
| Training objective | Cross entropy |
| Evaluation objective | Top-1 accuracy |
| | |
| Number of workers | 32 (unless otherwise specified) |
| Topology | Ring (unless otherwise specified) |
| Gossip weights | Metropolis-Hastings (1/3 for ring, $w_{ij} = 1/(\max(n_i, n_j) + 1)$, worker $i$ has $n_i$ direct neighbors) |
| Data distribution | Identical: workers can sample from the whole dataset |
| Sampling | With replacement (i.i.d. ), *no* shuffled passes |
| | |
| Batch size | 16 patches per worker |
| Momentum | 0.9 (heavy ball / PyTorch default) |
| Learning rate | Exponential grid or tuned for lowest training loss after 25 epochs |
| LR decay | Step-wise, $\times 0.1$ at epoch 75% and 90% of training |
| LR warmup | None |
| # Epochs | 100 (full training) or only 25 (initial phase), based on total number of gradient accesses across workers |
| Weight decay | $10^{-4}$ |
| Normalization scheme | no normalization layers |
| Exponential moving average | $\mathbf{x}_{\text{ema}}^{(t)} = 0.95 \mathbf{x}_{\text{ema}}^{(t-1)} + 0.05 \mathbf{x}^{(t)}$. This influences evaluation, not training |
| | |
| Repetitions per training | Just 1 per learning rate, but experiments are very consistent across similar learning rates |
| Reported metrics | *Loss after 2.5 k steps*: to reduce noise, we take two measures: (I) we use exponential moving average of the model parameters, and (II) we fit a parametric model $\log(l) = at + b$ to the 25 loss evaluations $(t, l)$ closest to $t = 2500$. We then evaluate this function at $t = 2500$. |

# F  Additional experiments

In the main paper, we have focussed on the training loss in the initial phase of training of Cifar-10. We do find that our findings there do correlate with test accuracy after a complete training with 100 epochs. Figure 10 shows the test accuracy as training progresses, for plots ordered by improving training loss after 2.5 k steps.

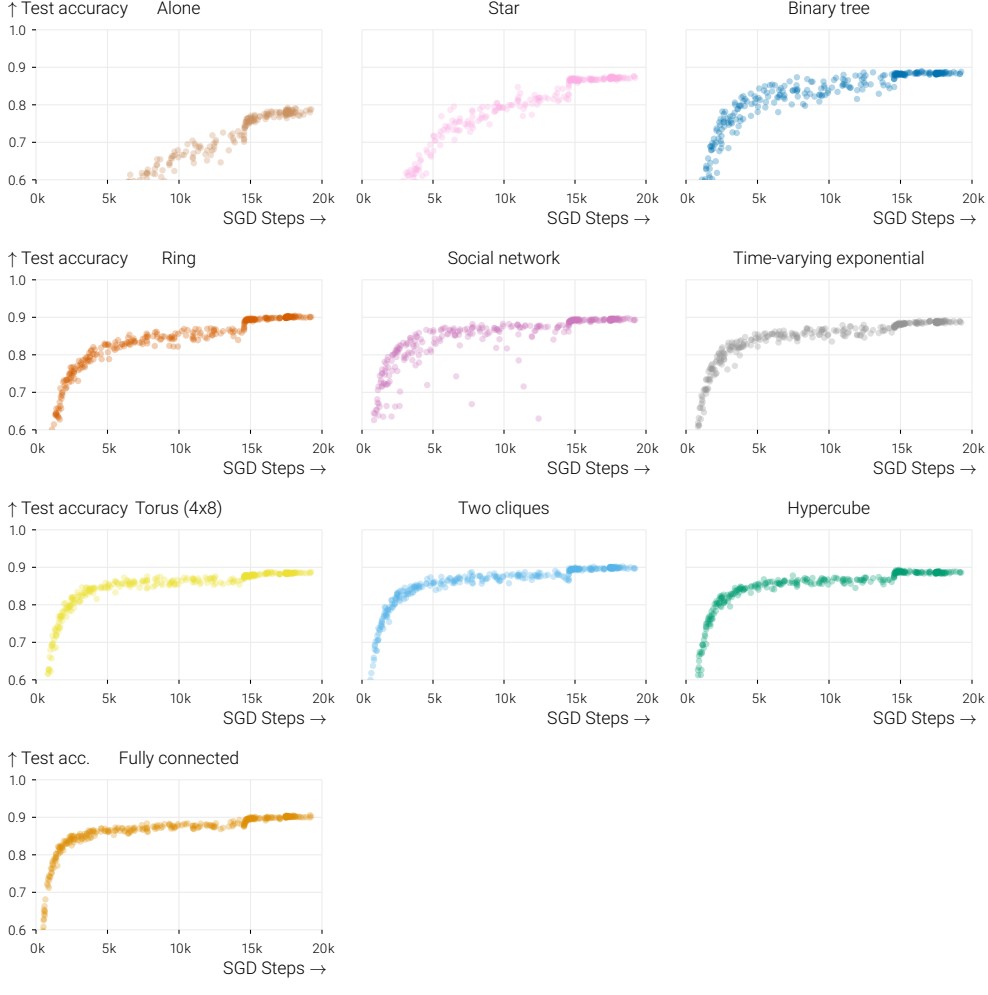

Figure 10: Test accuracy over the course of training a VGG-11 network on Cifar-10. See Appendix E for all details on the experimental setup. The plots are ordered by improving training loss after 2.5 k SGD steps. This ordering correlates well with the speed of improvements in test accuracy.

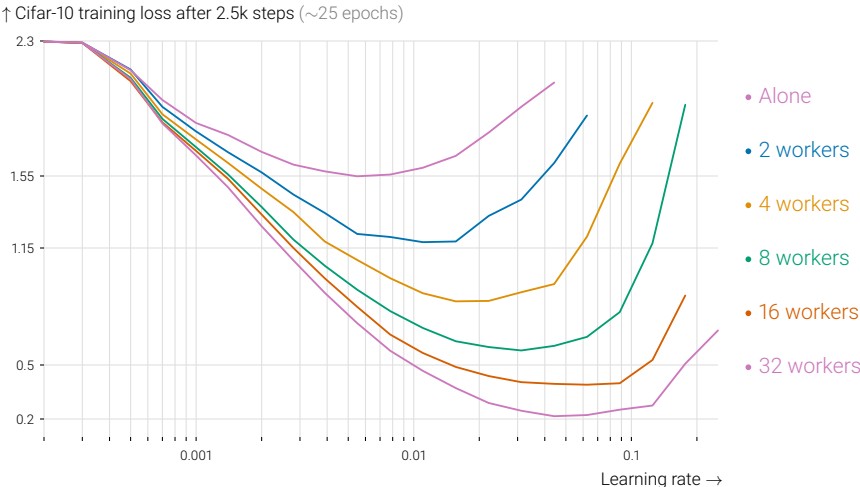

Figure 11: Training loss reached after 2.5 k SGD steps with fully-connected topologies of varying size. Averaging with more workers speeds up convergence for fixed learning rates, but also allows larger learning rates to be used. This plot serves as a reference for Figure 4, which shows similar plots for a variety of graph topologies.

## F.1 Results on Fashion MNIST

We replicated our main experiments (Cifar-10/VGG) on another dataset and another network architecture. We chose for the Fashion MNIST dataset [24] and a simple multi-layer perceptron architecture with one hidden layer of 5000 neurons and ReLU activations. We list the details of our experimental setup in Table 3. We varied two key parameters compared to our Cifar-10 results: we used 64 workers instead of 32, and used SGD *without* momentum and *without* weight decay. Because this task is easier than Cifar-10, the initial phase where both training and test loss converge at similar rates is shorter. We therefore consider the first 500 steps as the 'initial phase', as opposed to 2500.

Figures 12 and 13 correspond to figures 4 and 6 from the main paper. We find that the conclusions from the paper also hold in this different experimental setting.

Table 3: Experimental settings for Fashion MNIST. Differences with Cifar-10 in **red**.

| | |
|---|---|
| Dataset | **Fashion MNIST** [24] |
| Data augmentation | **None** |
| Data normalization | Subtract mean 0.2860 and divide standard deviation 0.3530 |
| Architecture | **MLP** ($28 \times 28 \rightarrow$ ReLU $\rightarrow 5000 \rightarrow$ ReLU $\rightarrow 10$) |
| Training objective | Cross entropy |
| Evaluation objective | Top-1 accuracy |
| Number of workers | **64** (unless otherwise specified) |
| Topology | Ring (unless otherwise specified) |
| Gossip weights | Metropolis-Hastings (1/3 for ring, $w_{ij} = 1/(\max(n_i, n_j) + 1)$, worker $i$ has $n_i$ direct neighbors) |
| Data distribution | Identical: workers can sample from the whole dataset |
| Sampling | With replacement (i.i.d. ), *no* shuffled passes |
| Batch size | 16 patches per worker |
| Momentum | **0.0** |
| Learning rate | Exponential grid or tuned for lowest training loss after **500 steps** |
| LR decay | None, in the initial phase of training |
| LR warmup | None |
| # Epochs | 500 steps |
| Weight decay | **0** |
| Normalization scheme | no normalization layers |
| Exponential moving average | $\mathbf{x}_{\text{ema}}^{(t)} = 0.95\mathbf{x}_{\text{ema}}^{(t-1)} + 0.05\mathbf{x}^{(t)}$. This influences evaluation, not training |
| Repetitions per training | Just 1 per learning rate, but experiments are very consistent across similar learning rates |
| Reported metrics | *Loss after 500 steps*: to reduce noise, we take two measures: (I) we use exponential moving average of the model parameters, and (II) we fit a parametric model $\log(l) = at + b$ to the 25 loss evaluations $(t, l)$ closest to $t = 500$. We then evaluate this function at $t = 500$. |

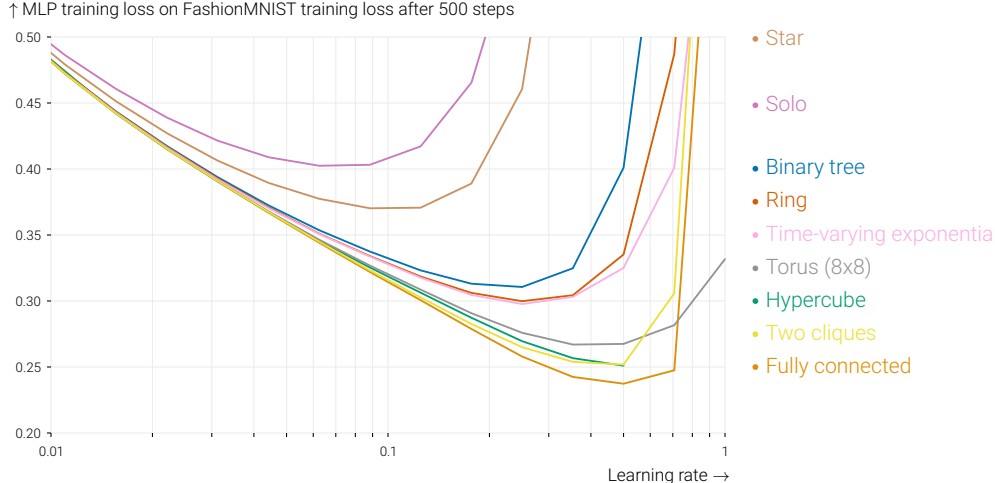

↑ MLP training loss on FashionMNIST training loss after 500 steps

- Star
- Solo
- Binary tree
- Ring
- Time-varying exponential
- Torus (8x8)
- Hypercube
- Two cliques
- Fully connected

Figure 12: Training loss reached after 500 SGD steps with a variety of 64-worker graph topologies. In all cases, averaging yields a small increase in speed for small learning rates, but a large gain over training alone comes from being able to increase the learning rate. While the star has a better spectral gap (0.0156) than the ring (0.0032), it performs worse, and does not allow large learning rates.

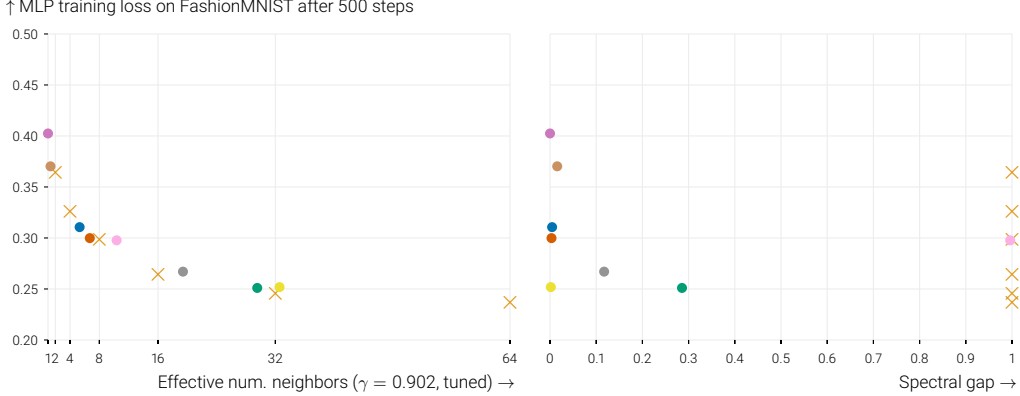

↑ MLP training loss on FashionMNIST after 500 steps

Figure 13: Fashion MNIST training loss after 500 steps for all studied topologies with their optimal learning rates. Colors match Figure 12, and × indicates fully-connected graphs with varying number of workers. After fitting a decay parameter $\gamma = 0.902$ that captures problem specifics, the effective number of neighbors (left) as measured by variance reduction in a random walk (like in Section 3) explains the relative performance of these graphs much better than the spectral gap of these topologies (right).

## F.2 Heterogeneous data

While our experimental and theoretical data only describe the setting in which workers optimize objectives with a shared optimum, we believe that our insights are meaningful for heterogeneous settings as well. With heterogeneous data, we observe two regimes: in the beginning of training, when the worker's distant optima are in a similar direction, everything behaves identical to the homogeneous setting. In this regime, our insights seem to apply directly. Heterogeneity only plays a role later during the training, when it leads to conflicting gradient directions. This behavior is illustrated on a toy problem in Figure 14. We run D-SGD on our isotropic quadratic toy problem ($d = 100$, $n = 32$), but where the optima are removed from zero as a normal distribution with standard deviations 0, $10^{-7}$, and $10^{-3}$ respectively. The (constant) learning rates are tuned for each topology in the homogeneous setting.

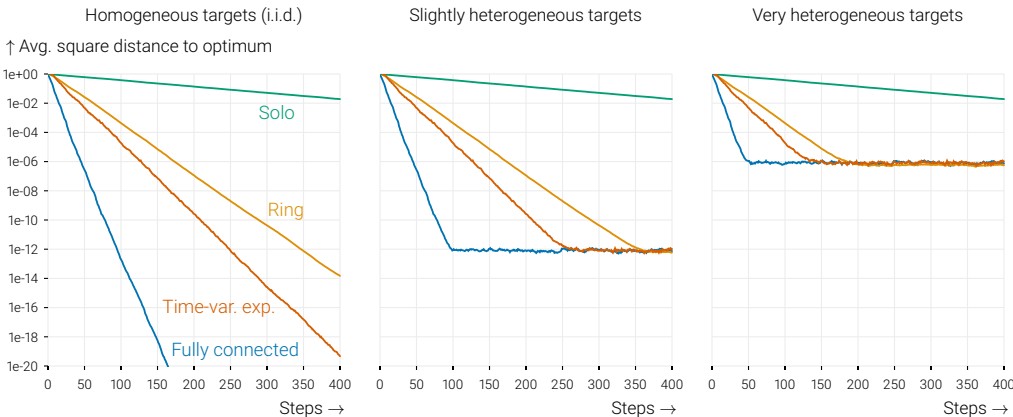

Figure 14: Convergence curves on our isotropic random quadratics problem (Section 3, with $d = 100$, $n = 32$), but where the optima are removed from zero as a zero-mean normal distribution with standard deviations $0$, $10^{-7}$, and $10^{-3}$ respectively. Constant learning rates are tuned independently for each topology in the homogeneous setting. Heterogeneity does not affect the initial phase of training, and our insights about maximum learning rates and the quality of communication topologies hold in this regime.

## F.3 The role of $\gamma$ in the experiments

In Figure 5, we optimize $\gamma$ independently for each topology, minimizing the Mean Squared Error between the normalized covariance matrix measured from checkpoints of Cifar-10 training and the covariance in a random walk with the decay parameter $\gamma$. The bottom two rows of Figure 15 below show how Figure 5 would change, if you used a $\gamma$ that is either much too low, or too high.

In Figure 6, we choose a value of $\gamma$ (shared between all topologies) that yields a good correspondence between the performance of fully connected topologies (with 2, 4, 8, 16 and 32 workers) and the other topologies. We opt for sharing a single $\gamma$ here, to test whether this metric could have predictive power for the quality of graphs. Figure 16 below shows how the figure changes if you use a value of $\gamma$ that is either much too low, or much too high.

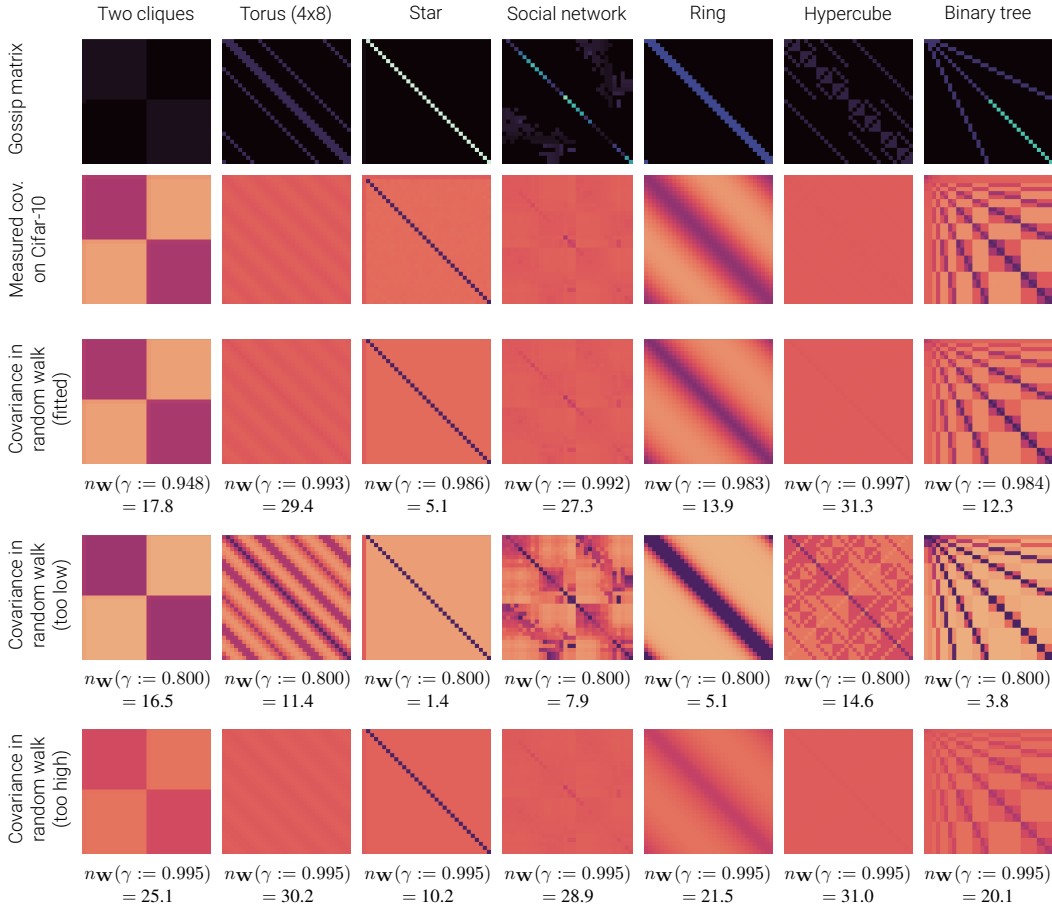

Figure 15: Extension of Figure 5. Measured covariance in Cifar-10 (second row) between workers using various graphs (top row). After 10 epochs, we store a checkpoint of the model and train repeatedly for 100 SGD steps, yielding 100 models for 32 workers. We show normalized covariance matrices between the workers. These are very well approximated by the covariance in the random walk process of Section 3 (third row). We print the fitted decay parameters and corresponding 'effective number of neighbors'. The bottom two rows show how Figure 5 would change, if you used a $\gamma$ that is either much too low, or too high.

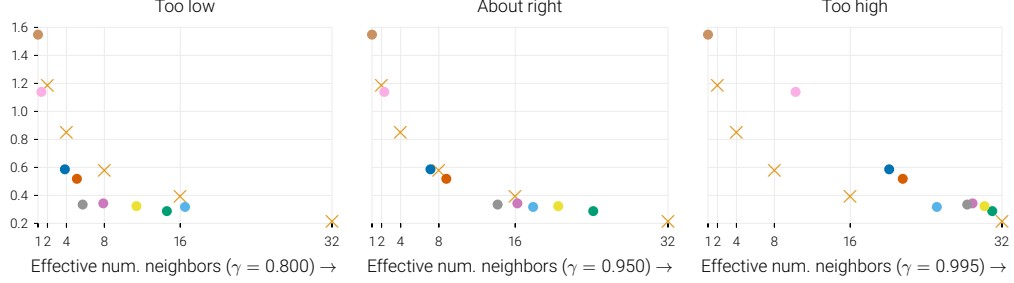

Figure 16: Extension of Figure 6, demonstrating how the fit changes if you use a value of $\gamma$ that is either too low (left) or too high (right).