# OpenReview forum: "Beyond spectral gap: the role of the topology in decentralized learning"
_NeurIPS.cc/2022/Conference — NeurIPS 2022 Accept_

### Official Review · Reviewer_nMZQ · 2022-06-25

**Rating:** 7
**Confidence:** 4
**Soundness:** 4 excellent
**Presentation:** 2 fair
**Contribution:** 4 excellent

**Summary:**

The paper analyzes the D-SGD algorithm where workers (nodes) in a graph collaboratively perform stochastic gradient descent of an objective function. While the existing convergence rates of D-SGD mostly depend on the spectral gap of the adjacency matrix (assuming small learning rate), this paper suggests an alternative analysis based on the "effective number of neighbors," which is defined through specific random walks on the communication graph.

In Section 3, the authors define the notion of effective number of neighbors $n_W(\gamma)$, which is roughly the variance reduction rate of random walks (with decay $\gamma$) on the graph $W$ compared to $W = I$. They derive a convergence rate depending on $n_W$, not the spectral gap of $W$, on a quadratic toy problem. They also extend this result to strongly convex functions. This analysis can explain the convergence of D-SGD even when the spectral gap is zero.

Finally, they verify their theory by training VGG-11 on CIFAR-10. They argue that $n_W$ is better aligned with empirical training performance than the spectral gap.

**Questions:**

- Could you provide a high-level motivation for introducing the decay $\gamma$? I understand a parameter is technically needed to fit the random walk model, but is there a good interpretation of it?
- Is it possible to compare the maximum learning rate in Theorem I with learning rate in other literatures?
- How exactly are the $\gamma$'s in the experiment chosen? From Figure 2, it seems that the ordering of $n_W$ of different graphs depends a lot on $\gamma$. Can you also show what happens to Figure 5&6 if we use different $\gamma$? Are those results still true?
- Is it possible to relax Assumption B (IV)? The toy problem of Section 3 does not satisfy it since the spectrum of $\mathbf{d} \mathbf{d}^\top$ can be arbitrary for $\mathbf{d} \sim \mathcal{N}^d(0, 1)$. Maybe some sort of average case definition is needed (just like Definition C)?




**Limitations:**

The authors have well addressed the limitations of their work.

**Strengths And Weaknesses:**

Strengths
- The paper captured a weakness of current theories on D-SGD and provided an improved explanation on its convergence. Now, their theory can explain the convergence when the spectral gap is zero and also better align with empirical performance of neural networks trained with D-SGD.
- The paper introduced an interesting random walk model on a graph as a proxy for the training dynamics of D-SGD. Based on this, they define a novel concept of $n_W$, which is a key in their analysis.
- The paper performed exhaustive experiments on various types of graph (including time-varying topology) to confirm the theory.

Weaknesses
- I believe the writing of the paper can be much improved. There were notations (e.g. $\zeta$, $\bar{\mathbf{P}}$) that are used before defined, so I had to go between the main text and appendix back and forth.
- In line 149, the paper is giving the rate in terms of $n_W$, which is not yet defined. I think it is logically better to move the result after Section 3.2 (after $n_W$ is defined).
- There were many typos and some of them slowed down my reading significantly.

Typos
- line 193, page 6: $\zeta = d + 2$ instead of $\zeta = d + 1$
- displayed equation below line 471, page 17: $\gamma^{\frac{k - 1}{2}}$ instead of $\gamma^{\frac{k}{2}}$ (think about $\mathbf{z}^{(1)}$)
- displayed equation below line 473, page 17: $\gamma^{k - 1}$ instead of $\gamma^k$ (same reason)
- line 476, page 17: assumptions of "Lemma" 2
- proof of Lemma 4, page 18: Missing subscript of $\mathbf{b}^{(t)}$. Result is correct, but the proof has a wrong scaling. The second line leads to $\gamma (1 - r) \mathbf{b}_{ni + j}^{(t)} + (1 - r)^t$ -- should be fixed.
- line 501, page 18: the inequality is in the wrong direction. Should be $\mathbf{b}^{(t + 1)} \geq ...$.
- line 521, page 19: $\nabla h(x)$
- Figure 13, page 27: the figure and caption say different $\gamma$

---

> ### Author Response · Authors · 2022-08-02
> **Reply**
>
> Dear Reviewer,
>
> Thank you for your very thorough review. Your suggestions on clarity and typos are much appreciated, and we will incorporate them into the revised paper. We hope to answer your questions below:
>
> __Intuition for $\gamma$.__ The decay parameter $\gamma$ modulates the sensitivity to communication delays. As $\gamma$ approaches 1, it does not matter how old an update is. Even updates that travel many hops get approximately the same weight as updates from a node itself. This is fitting when the learning rate is small compared to the smoothness of the function. On the other end of the spectrum, $\gamma=0$ indicates that delayed updates are useless, and only the updates from directly connected neighbors are beneficial. A small $\gamma$ is applicable when the learning rate is very large.
>
> __Learning rate in Theorem 1.__ Standard D-SGD rates [8] rely on step-sizes of order $O(\min(1/T, 1 - \lambda_2(W))$, which are much smaller than $n_{W(\gamma)} / \zeta$ since the number of iterations $T$ is generally much larger than $\zeta$, and the spectral gap of the network might be very small.
>
> With these small step-sizes, [8] obtain an effective batch size of $n$ (and so divide the residual variance by a corresponding factor).
> Instead, we show improvement for any (large) constant step-size, but the effective batch size is “only” $n_{W(\gamma)}$, which is natural since linear speedup in $n$ is not achievable for large step-sizes and sparse networks. Corollary II also provides a simple comparison: D-SGD is comparable to mini-batch SGD, where the effective batch size depends on the connectivity of the network and the learning rate.
>
> Thank you for pointing this out, and we will try to make it clearer in a revision of this paper.
>
> __Choice of $\gamma$.__  In Figure 5, we optimize $\gamma$ independently for each topology, minimizing the Mean Squared Error between the normalized covariance matrix measured from checkpoints of Cifar-10 training and the covariance in a random walk with the decay parameter $\gamma$. The new Figure 15 (Appendix) shows how Figure 5 would change if you used a $\gamma$ that is either much too low, or too high.
>
> In Figure 6, we choose a value of $\gamma$ (shared between all topologies) that yields a good correspondence between the performance of fully connected topologies (with 2, 4, 8, 16 and 32 workers) and the other topologies. We opt for sharing a single $\gamma$ here, to test whether this metric could have predictive power for the quality of graphs. The new Figure 16 (Appendix) shows how the figure changes if you use a value of $\gamma$ that is either much too low, or much too high.
>
> __Toy problem and Assumption B.__  We stated Assumption B (IV) in this form for simplicity, but it can be relaxed by asking directly that $\mathbb{E} \| \nabla f_{\xi,i}(x^{(t)}) - \nabla f_{\xi,i}(x^\star)\|^2 \leq 2 \zeta D_f(x^\star, x^{(t)})$ (see Appendix D.2, between Equations (22) and (23), which can also be implied by assuming that each $f_\xi$ is $\zeta_\xi$-smooth, with $\mathbb{E} \left[\zeta_\xi D_{f_\xi}(x^\star, x^{(t)})\right] \leq \zeta D_f(x^\star, x^{(t)})$. These weaker forms would be satisfied by the toy problem of Section 3.

---

> > ### Comment · Reviewer_nMZQ · 2022-08-03
> > **Thanks for the response**
> >
> > I appreciate the detailed answers including the additional experiments on the choice of $\gamma$ (Figure 15, 16). Now it is more clear that the trends observed in the paper still exist to some extent even when different $\gamma$ is used. While I am generally happy with the paper and leaning to acceptance, I am very concerned with the readability of the paper.
> >
> > I don't know if a revision is allowed during the discussion period, but it would be great if the authors can incorporate the points I raised in the review, as well as some parts of their own answers. I believe this will improve the readability a lot.

---

> > > ### Author Response · Authors · 2022-08-03
> > > **Revision**
> > >
> > > Thank you for your quick reply.
> > > We agree with all your concrete suggestions on clarity and typos, and are very grateful for your in-depth review and contributions to the quality of the paper.
> > >
> > > For the initial rebuttal, we had already
> > > - corrected the typo on line 193,
> > > - corrected all typos you found in the appendix.
> > >
> > > We just uploaded a new revision, in which we
> > > - removed the symbol $\bar P$,
> > > - clearly defined $\zeta$ from the start,
> > > - reorganized Section 3 so the definition of $n_W$ appears earlier,
> > > - provided more intuition for $\gamma$,
> > > - added a small discussion that compares the learning rates in Theorem 1 to standard learning rates,
> > > - incorporated feedback from the other reviewers.
> > >
> > > We would be happy to integrate any other concrete suggestions on clarity in the camera ready version.

---

### Official Review · Reviewer_9GNP · 2022-07-06

**Rating:** 6
**Confidence:** 5
**Soundness:** 3 good
**Presentation:** 4 excellent
**Contribution:** 2 fair

**Summary:**

In this paper, the authors showed that decentralized learning allows a larger learning rate compared to centralized learning, which can accelerate the training process. The effective number of neighbors is defined in the paper which measures the ratio of the asymptotic variance of the iterations. As a main result, the authors provided the theorem about the larges learning rate that gives the best convergence guarantees based on the local neighborhood size. The authors also provided the results of the experiment based on the CIFAR10 dataset.

**Questions:**

(1) Same as the second point in weaknesses, is there any theoretical direction for the non-convex problems. It's maybe hard to get the same results as the convex case, but there is possible to get some upper bound or lower bound for the learning rate.
(2) In the theory part, the authors use randomized D-SGD defined in equations (4). However, in practice, the standard D-SGD is used widely, which will average the parameters of neighbors and update the parameters every step without probability. It would be better to extend the current theory to the standard D-SGD.
(3) For disconnected cases, it is more reasonable to look at batch size = batch size of decentralized case * num of workers rather than use the same batch size as the decentralized case. In the current setting, the larger stepsize mainly comes from a larger batch size compared to the decentralized case. If we use batch size * num of workers, would these results still hold?

**Limitations:**

Yes, the authors have adequately addressed the limitations and potential negative societal impact of their work.

**Strengths And Weaknesses:**

Strengths:
(1) Equations are written in a clear and reader-friendly way.
(2) Visualizations are useful to convey the authors' main idea.
(3) Paper is well-written and easy to follow.

Weaknesses:
(1) The related work section should discuss more related works.
(2) Although the theory of this paper is solid, it would be better to give some theoretical direction for the non-convex problems. Since data-parallel optimization is widely used in deep learning regimes rather than strongly convex problems.

---

> ### Author Response · Authors · 2022-08-02
> **Reply**
>
> Dear Reviewer,
>
> Thank you for your review and valuable feedback. We answer your questions below:
>
> __Randomized D-SGD.__  We can obtain similar (slightly better) results for the deterministic alternating update rule compared to randomized D-SGD. The new Appendix D.4 outlines how to obtain these results. Empirically, we confirmed that the typical alternating algorithm performs consistently slightly better than the randomized version, but that the relative performance of topologies does not change between the algorithms.
>
> __Non-convex theory.__ While our experiments show that the convex theory is already useful for deep learning, we agree that the non-convex setting is interesting. Although the core ideas of our analysis (semi-local analysis with matrix $M$) would remain the same, some details would change. In particular:
>
> - The measure of suboptimality would switch from $\|x_t - x_\star\|^2_M$ to $\|\nabla f(x_t)\|^2_M$.
> - The definition of noise should be adapted from assuming that each stochastic function is $\zeta$-smooth to assuming a bound of the following type for all $f_\xi$ : $\| \nabla f_\xi (x_t)\|^2 \leq (\zeta / L) \| \nabla f(x_t)\|^2 + \sigma^2$, where $f$ is $L$-smooth.
>
> We conjecture that, under these assumptions, D-SGD is comparable to mini-batch SGD with some batch-size that depends on the connectivity of the network and the step-size. Yet, completely adapting the full proof remains non-trivial. One key difficulty is that performing gossip steps might increase the function value ($f(Wx) \geq f(x)$), and correctly bounding the gap might require new assumptions (such as Lipschitzness of $f$, or some specific initialization).
>
> __Batch size.__  The baseline you request is already included in our results. In the setting we study, the *disconnected* topology with batch size $b \times n$ (per worker) is equivalent to the *fully connected* topology with batch size $b$ per worker. As you said, in the i.i.d. setting, the benefit of averaging is variance reduction. Lower variance implies a larger ‘effective batch size’. The paper offers a way to reason about *how much* each topology can reduce the variance compared to training alone, while keeping the local computation cost fixed to a batch size of $b$. The fully-connected case you described (effective batch size $b \times n$) corresponds to the best-achievable variance reduction for a given batch size $b$.
>
> __Related work.__  We believe our related work section covers the most relevant prior work. Since the submission of this paper, we were made aware of the following works:
> - [A], which shows that D-SGD adds some implicit regularization (a different benefit than the large learning rate), and attains optimal statistical rates. Yet, their optimization error bound is looser than ours, and in particular relies on the spectral gap.
> - [B] also show that larger (constant) step-sizes can be used in decentralized settings, but their analysis focuses on Decentralized Kernel Regression, does not cover stochastic gradient updates, and relies on statistical concentration of local objectives rather than analysis on local neighborhoods.
>
> If you have particular papers or areas in mind that we missed, we would be very grateful if you could share a reference, so we can discuss it further.
>
> ---
>
> [A] Richards D. Graph-dependent implicit regularisation for distributed stochastic subgradient descent. Journal of Machine Learning Research. 2020.
>
> [B] Richards D, Rebeschini P. Optimal statistical rates for decentralised non-parametric regression with linear speed-up. Advances in Neural Information Processing Systems. 2019;32.

---

> > ### Comment · Reviewer_9GNP · 2022-08-04
> > **Thanks for the responses**
> >
> > Thanks for your detailed responses, which have answered most of my questions.

---

### Official Review · Reviewer_WZhq · 2022-07-11

**Rating:** 6
**Confidence:** 4
**Soundness:** 3 good
**Presentation:** 4 excellent
**Contribution:** 3 good

**Summary:**

This paper deeply investigated the role of topology in decentralized learning. Instead of the commonly adopted spectral gap in the literature, it proposed the "effective number of neighbors" concept and further revealed that the benefits of "good" topology are enabling larger learning rates so that it speeds up optimizations. The paper provides a sound proof of the convex case and observed similar behavior in deep learning.

**Questions:**

1.  The D-SGD update adopted the uncommonly used probability update rule, i.e. execute the local gradient update with some probability and gossip communication with the rest case. It makes the analysis simplified since the expectation of Lyapunov becomes the convex combination of two steps independently instead of coupling them together. What about the case that one local gradient always followed by one gossip communication? How to modify the proof framework to cover this case?

2. Can we rewrite $M = (1-r)\sum_{k=1}^\infty\gamma^{k-1}W^{2k}$ into the form $M = (1-\gamma) W^{2}(I-\gamma W^2)^{-1}$ (assuming the inverse exists), since it looks closer to toy example Eq. (3). Also, personally feel the Definition I (effective number of neighbors) in the appendix is better moved to the main context so that the reader is easier to understand what the intuition of effective neighbors means.

3. The toy example use an isotropic quadratic function $\mathbb{E} \frac{1}{2}(d^Tx)^2 = \frac{1}{2}\\|x\\|^2$, which has a special property that the gradient noise at the optimal point $w^\star$ vanish so that the linear speedup is established. It is more common to use Least-Mean-Square as the toy example, which is closer to the later theorem form.

**Limitations:**

The nice detailed analysis is based on homogenous data and all workers shared the same loss function. I don't think heterogonous data or different minimizer cases should be covered by the theorem in this paper. But it will be nice to use some experiments to show the sensitivity of the conclusion when that assumption is not valid.

**Strengths And Weaknesses:**

Originality: The role of the topology can influence the step-size choice in decentralized learning is not a surprisingly new observation, but this is the first time I saw a well-quantified way to describe it and show it in rigorous proof.
Quality & Clarity: The presentation is clear and well organized. I can follow the logic and proof easily. The proof looks sound to me.
Significance: The work is a theoretical explanation work, which can be quite useful. However, the conclusion and the proof technique seem to rely on several strict assumptions. Not sure how easy to extend this framework to more general or complicated settings.

---

> ### Author Response · Authors · 2022-08-02
> **Reply**
>
> Dear Reviewer,
>
> Thank you for your review and your insightful comments. We answer your questions below:
>
> __Randomized v.s. alternating update rule.__  We can obtain similar (slightly better) results for the alternating update rule compared to randomized D-SGD. The new Appendix D.4 outlines how to obtain these results. Empirically, we confirmed that the typical alternating algorithm performs consistently slightly better than the randomized version, but that the relative performance of topologies does not change between the algorithms.
>
> __Matrix M: Eqs. (5) and (3).__  Thank you for this suggestion. Yes, these are equivalent, and we will clarify this connection.
>
> __Definition I (effective num. neighbors).__  The current Definition A in the main paper is an informal (“in words”) equivalent of Definition I. But we agree that the main text could benefit from the more explicit form in Definition I.
>
> __Toy function.__  We chose the isotropic quadratic toy problem because we can easily derive exact linear rates for this problem, and establish an obvious link with the notion of “effective number of neighbors”. We believe that the simplicity of this toy problem is a useful pedagogical complement to the more general theory in Section 4.
>
> __Heterogeneous data__.  With heterogeneous data, we observe two regimes: in the beginning of training, when the worker's distant optima are in a similar direction, everything behaves identical to the homogeneous setting. In this regime, our insights on optimal learning rates and the quality of communication graphs are applicable. Heterogeneity seems to only play a role later during the training, when it leads to conflicting gradient directions. This behavior is illustrated the new Figure 14 (appendix), where we run D-SGD on our isotropic quadratic toy problem, but where the worker's optima are removed from zero by a zero-mean normal distribution with varying standard deviation.

---

### Meta-Review · Area_Chair_vS9H · 2022-08-22

**Recommendation:** Accept
**Confidence:** Less certain

**Metareview:**

The paper studies decentralized optimization and considers all machines work on the data that follow the same distribution. Most of the reviewers think the paper is interesting. I recommend an acceptance.

**Award:**

No

---

### Decision · Program_Chairs · 2022-09-14

Accept